# MIP-1α Expression Induced by Co-Stimulation of Human Monocytic Cells with Palmitate and TNF-α Involves the TLR4-IRF3 Pathway and Is Amplified by Oxidative Stress

**DOI:** 10.3390/cells9081799

**Published:** 2020-07-29

**Authors:** Sardar Sindhu, Nadeem Akhter, Ajit Wilson, Reeby Thomas, Hossein Arefanian, Ashraf Al Madhoun, Fahd Al-Mulla, Rasheed Ahmad

**Affiliations:** 1Animal & Imaging Core Facility, Dasman Diabetes Institute (DDI), Al-Soor Street, P.O. Box 1180, Dasman 15462, Kuwait; ashraf.madhoun@dasmaninstitute.org; 2Immunology & Microbiology, Dasman Diabetes Institute (DDI), Al-Soor Street, P.O. Box 1180, Dasman 15462, Kuwait; nadeem.akhter@dasmaninstitute.org (N.A.); ajit.wilson@dasmaninstitute.org (A.W.); reeby.thomas@dasmaninstitute.org (R.T.); hossein.arefanian@dasmaninstitute.org (H.A.); 3Genetics & Bioinformatics, Dasman Diabetes Institute (DDI), Al-Soor Street, P.O. Box 1180, Dasman 15462, Kuwait; fahd.almulla@dasmaninstitute.org

**Keywords:** palmitate, TNF-α, oxidative stress, ROS, MIP-1α, TLR4, IRF3, metabolic inflammation

## Abstract

Metabolic inflammation is associated with increased expression of saturated free fatty acids, proinflammatory cytokines, chemokines, and adipose oxidative stress. Macrophage inflammatory protein (MIP)-1α recruits the inflammatory cells such as monocytes, macrophages, and neutrophils in the adipose tissue; however, the mechanisms promoting the MIP-1α expression remain unclear. We hypothesized that MIP-1α co-induced by palmitate and tumor necrosis factor (TNF)-α in monocytic cells/macrophages could be further enhanced in the presence of reactive oxygen species (ROS)-mediated oxidative stress. To investigate this, THP-1 monocytic cells and primary human macrophages were co-stimulated with palmitate and TNF-α and mRNA and protein levels of MIP-1α were measured by using quantitative reverse transcription, polymerase chain reaction (qRT-PCR) and commercial enzyme-linked immunosorbent assays (ELISA), respectively. The cognate receptor of palmitate, toll-like receptor (TLR)-4, was blunted by genetic ablation, neutralization, and chemical inhibition. The involvement of TLR4-downstream pathways, interferon regulatory factor (IRF)-3 or myeloid differentiation (MyD)-88 factor, was determined using IRF3-siRNA or MyD88-deficient cells. Oxidative stress was induced in cells by hydrogen peroxide (H_2_O_2_) treatment and ROS induction was measured by dichloro-dihydro-fluorescein diacetate (DCFH-DA) assay. The data show that MIP-1α gene/protein expression was upregulated in cells co-stimulated with palmitate/TNF-α compared to those stimulated with either palmitate or TNF-α (*P* < 0.05). Further, TLR4-IRF3 pathway was implicated in the cooperative induction of MIP-1α in THP-1 cells, and this cooperativity between palmitate and TNF-α was clathrin-dependent and also required signaling through c-Jun and nuclear factor kappa-light-chain-enhancer of activated B cells (NF-κB). Notably, ROS itself induced MIP-1α and could further promote MIP-1α secretion together with palmitate and TNF-α. In conclusion, palmitate and TNF-α co-induce MIP-1α in human monocytic cells via the TLR4-IRF3 pathway and signaling involving c-Jun/NF-κB. Importantly, oxidative stress leads to ROS-driven MIP-1α amplification, which may have significance for metabolic inflammation.

## 1. Introduction

Metabolic inflammation in obesity is associated with multiple factors such as oxidative stress, proinflammatory cytokines/chemokines, adipokines, and free fatty acids. The co-expression of these factors may damage the cellular structures and impair signaling, leading to obesity-related complications such as insulin resistance, metabolic syndrome, and liver and cardiovascular diseases [1]. Activated monocytes and M1 macrophages secrete proinflammatory cytokines and chemokines including tumor necrosis factor (TNF)-α, IL-1β, IL-6, and α/β chemokines [2]. The elevated levels of these factors may induce or promote the expression of reactive oxygen species (ROS) and reactive nitrogen species (RNS) in monocytes and macrophages [3]. Through a feedback loop mechanism, ROS or RNS may, in turn, induce the proinflammatory cytokines, adhesion molecules, and growth factors, and this crosstalk between ROS and proinflammatory factors plays a key role in sustaining a state of chronic low-grade inflammation [4].

The chemotactic cytokines called macrophage inflammatory proteins (MIPs) include MIP-1α (also called CCL3) and MIP-1β (also called CCL4), which orchestrate immune responses to infection and inflammation [5]. MIPs are important players in the induction of other proinflammatory cytokines such as TNF-α, IL-1β, and IL-6 in activated macrophages and fibroblasts; while, through their stimulatory effects on granulocytes, MIPs can induce acute neutrophilic inflammation [6]. MIP-1α is a major chemoattractant for monocytes/macrophages, T-lymphocytes and neutrophils [7]. It is produced by several activated cells including monocytes, macrophages, CD8^+^ T lymphocytes, dendritic cells, endothelial cells, fibroblasts, and microglial cells [8]. MIP-1α is involved in the pathogenesis of many chronic inflammatory conditions such as rheumatoid arthritis, multiple myeloma, leukemogenesis, periodontitis, and Sjögren syndrome [9,10].

Chemokines can be expressed in both immune and non-immune cells by a wide variety of inflammatory stimuli involving TNF-α, IL-1β, complement fraction C5a, viruses, bacterial LPS, leukotriene-B4, and free fatty acids (FFAs) [11]. Elevated circulatory levels of FFAs play a key role in inducing metabolic inflammation and insulin resistance [12]. FFAs may activate inflammatory signaling by binding to specific toll-like receptors (TLRs) [13]. TLR4 is a member of the pattern-recognition receptors family and is activated by pathogen-associated molecular patterns that include lipids, proteins, lipoproteins, LPS, and alarmins such as high-mobility group box-1 protein [14]. Ligand binding to TLR4 activates the downstream signaling via two different pathways i.e., MyD88-dependent pathway or TIR-domain-containing adapter-inducing interferon-β (TRIF)/IRF3-dependent pathway [15]. In the “classical or canonical” MyD88-dependent pathway, TLR recruits TIR domain-containing adaptor protein in cell membrane recruits MyD88, triggering the early-phase NF-κB activation. In the TRIF-dependent canonical pathway, TLR4 forms a signaling complex with TRIF related adaptor molecule (TRAM) and TRIF, resulting in the late-phase NF-κB activation via IRF3 engagement and IFN-β transcription. IKK phosphorylates IκBα and IκBβ, leading to polyubiquitination and degradation of IκBs via 26S proteasome and release of NF-κB, which translocates to the nucleus to activate target gene expression [16,17]. In the “alternative or non-canonical pathway” mainly involved in immune response, TNF receptor (TNFR) activation initiates the NF-κB-inducing kinase (NIK) to stimulate IKKα-induced phosphorylation and processing of NF-κB2 precursor. Activated NF-κB2, NIK, and RelB form a complex that translocates to the nucleus to activate target gene expression [18,19]. Given the emerging role of innate immune TLR4 receptor in nutrient sensing and metabolic inflammation, it is important to study which TLR4-downstream mechanism(s) can lead to MIP-1α expression in human monocytic cells and/or macrophages.

Palmitate is a long-chain saturated fatty acid that activates TLR4-mediated signaling and induces the expression of TNF-α, IL-6, and IL-1β in immune cells [20] and MIP-1α expression in human monocytic cells [21]. TNF-α is a critical proinflammatory cytokine associated with metabolic inflammation and insulin resistance [22]. Besides proinflammatory cytokines and chemokines such as MIP-1α [23], hypoxia and oxidative stress are also known to induce or exacerbate metabolic inflammation [24,25]. Given that both palmitate and TNF-α have been associated with metabolic inflammation, we asked if these two agents could cooperatively induce MIP-1α expression in human monocytic cells/macrophages and whether the oxidative stress per se could induce MIP-1α production. Herein, we show that palmitate and TNF-α co-induce MIP-1α expression in THP-1 monocytic cells or macrophages through the mechanism involving TLR4–IRF3 pathway, clathrin-mediated endocytosis (CME), and c-Jun/NFκB signaling, while the ROS also induces/increases MIP-1α secretion.

## 2. Materials and Methods

### 2.1. Cell Cultures and Stimulations

Human monocytic THP-1 cell line was purchased from the American Type Culture Collection (ATCC; Manassas, VA, USA). The cells were incubated at 37 °C (5% CO_2_ with humidity) in RPMI-1640 cell culture medium (Gibco, Life Technologies, Carlsbad, CA, USA) containing 10% FBS (Cat. #16140-071, Gibco, Life Technologies), 2 mM L-glutamine (Cat. #25030-081, Gibco, Invitrogen, Waltham, MA, USA), 1 mM sodium pyruvate, 10 mM HEPES, 50 U/mL penicillin & 50 μg/mL streptomycin (Cat. #15140-122, Gibco, Invitrogen), 0.05 mM 2-mercaptoethanol (Cat. #M3148, Sigma-Aldrich, St. Louis, MO, USA), and 100 μg/mL Normocin (Cat. #ant-nr, Gibco, Invitrogen). THP-1 cells defective in MyD88 activity known as THP1-XBlue™-defMyD or MyD88^−/−^ cells were purchased from InvivoGen (San Diego, CA, USA). THP1-XBlue™-defMyD cells were incubated in RPMI-1640 complete cell culture medium containing Zeocin (200 μg/mL) and HygroGold (100 μg/mL; InvivoGen). Before stimulation, THP-1 cells were transferred to normal medium and plated at a cell density of 10^6^ cells/mL/well in triplicate wells of 12-well plates (Costar, Corning Incorporated, Tewksbury, MA, USA). Cells were stimulated with palmitate (200 μM; Cat. #P5585, Sigma-Aldrich) and/or recombinant human TNF-α (10 ng/mL; Cat. #210-TA-100, R&D Systems, Minneapolis, MN, USA) or 0.1% fatty acid-free bovine serum albumin (BSA; Probumin® Cat. #82-02-004, Millipore-Sigma, Burlington, MA, USA) and incubated at 37 °C for 24 h. Cells were harvested for RNA isolation and culture supernatants were collected for measuring levels of secreted MIP-1α protein.

### 2.2. Isolation of Human Peripheral Blood Mononuclear Cells (PBMC), Differentiation, and Stimulation of Primary Macrophages

The peripheral blood from three healthy adult donors, 40 mL each, was collected in EDTA vacutainer tubes at phlebotomy unit of the Dasman Diabetes Institute (Kuwait City, Kuwait). All blood donors gave informed written consent before sample collection for the study. The collection and processing of blood samples was in accordance with the Declaration of Helsinki 1975 (revised in 2013) and the protocol was approved by the Ethics Committee of Dasman Diabetes Institute (Ref. #RA-2014-016, RA-AH-2016-007, & RA-2015-027). PBMC were isolated using standard HistoPaque density gradient technique as described elsewhere [26]. The cells were dispensed in triplicate wells of six-well plates (Costar, Corning Incorporated) at a cell density of 3 × 10^6^ cells/3 mL/well and cultured in serum-free medium at 37 °C for 3 h. The non-adherent cells were removed by gentle flushing with fresh serum-free medium and the adherent cells were further incubated at 37 °C for 24 h in RPMI complete medium containing 2% FBS. Cells were stimulated with palmitate (200 μM) and/or TNF-α (10 ng/mL) or 0.1% BSA and incubated at 37 °C for 24 h. At the end of incubation, cells were harvested for total RNA extraction, and supernatants were collected for measuring concentrations of MIP-1α secreted protein.

### 2.3. siRNA-Mediated Genetic Suppression of TLR4 and IRF3

THP-1 cells were washed and resuspended in 100 μL nucleofector solution (Amaxa Nucleofector^TM^ Kit V, Lonza Bioscience, Cologne, Germany). Cells were transfected separately with siRNAs (30 nM) against TLR4, IRF3, or scrambled (OriGene Technologies Inc. Rockville, MD, USA) and pmaxGFP (0.5 μg; Amaxa Nucleofector^TM^ Kit V, Lonza Bioscience, Cologne, Germany) using Amaxa Electroporation System (Amaxa Inc., Cologne, Germany) and following the manufacturer’s instructions. At 36 h post-transfection, THP-1 cells (10^6^ cells/mL/well) were treated in triplicate wells of 12-well plates with palmitate (200 μM) and/or TNF-α (10 ng/mL) or 0.1% BSA and incubated at 37 °C for 24 h. Cells were harvested for total RNA extraction and supernatants were collected for measuring MIP-1α secreted protein levels. The efficiency of siRNA transfection was assessed by fluorescence microscopy and target gene suppression was determined by real-time quantitative reverse transcription polymerase chain reaction (qRT-PCR).

### 2.4. TLR4 Neutralization

THP-1 monocytic cells seeded in triplicate wells of 12-well plates at a cell density of 10^6^ cells/mL/well were incubated at 37 °C for 40 min with anti-TLR4 neutralizing mAb (IgA2, 2 μg/mL; Cat. #mab2-htlr4, InvivoGen) or isotype-matched control mAb (IgA2, 2 μg/mL; Cat. #mabg1-ctrlm, InvivoGen). Later, antibody-labeled cells were treated with palmitate (200 μM) and/or TNF-α (10 ng/mL) or 0.1% BSA and incubated at 37 °C for 24 h. Cells were harvested for total RNA extraction and supernatants were collected for measuring MIP-1α secreted protein concentrations.

### 2.5. Chemical Inhibition of TLR4-Mediated Signaling

Oxidized 1-palmitoyl-2-arachidonoyl-sn-glycero-3-phosphocholine (OxPAPC) inhibits the TLR2- and TLR4-mediated signaling. It acts by competing with accessory proteins such as CD14, LBP, and MD2 that interact with bacterial lipids and inhibits the TLR2/4 mediated signaling [27]. To investigate the involvement of TLR4, THP-1 cells cultured in triplicate wells of 12-well plates at a cell density of 10^6^ cells/mL/well were treated with palmitate (200 μM) and/or TNF-α (10 ng/mL) or 0.1% BSA, with or without OxPAPC (30 μg/mL; Cat. #tlrl-oxp1, InvivoGen) and cells were incubated at 37 °C for 24 h. Later, cells were harvested for total RNA extraction and supernatants were collected for measuring MIP-1α secreted protein.

### 2.6. Trafficking Inhibition

To study trafficking through the CME pathway, THP-1 monocytic cells cultured in triplicate wells of 12-well plates at a cell density of 10^6^ cells/mL/well were treated (37 °C for 1 h) with a CME trafficking inhibitor chlorpromazine (CPZ; 20 μM) or with vehicle only [28]. Later, cells were treated with palmitate (200 μM) and/or TNF-α (10 ng/mL) or 0.1% BSA and incubated at 37 °C for 24 h to allow internalization via CME. At the end of incubation, cells were harvested for total RNA extraction and supernatants were collected for measuring levels of MIP-1α secreted protein.

### 2.7. Induction of Oxidative Stress, ROS Measurement, and Cell Treatments with Anti-Oxidants/ROS Scavengers

THP-1 cells cultured in triplicate wells of 12-well plates at a cell density of 10^6^ cells/mL/well were incubated for 2 h at 37 °C (5% CO_2_ with humidity) for conditioning. Cells were then incubated at 37 °C for 10 h with 10 mM hydrogen peroxide (H_2_O_2_) (9.8M, Cat. No. 822287.1000, Merck, Temecula, CA, USA) for induction of ROS-mediated oxidative stress [29]. Cell viability was assessed by trypan blue dye exclusion test and was found to be >87%, differing non-significantly from the viability of vehicle-treated control (Appendix A). The induction of ROS following treatment with H_2_O_2_ was determined using dichloro-dihydro-fluorescein diacetate (DCFH-DA) assay (Cat. #KP-06-003 BQC Kit, BioQueChem Inc., Llanera–Asturias, Spain), which measures the uptake of cell permeant 2′-7′dichlorofluorescein diacetate (DCFH-DA) fluorogenic probe. After incubation of cells with the probe for 15 min, DCFH-DA is hydrolyzed by cellular esterases into DCFH carboxylate anion, which is oxidized by ROS to yield a fluorescent product called 2′-7′dichlorofluorescein (DCF), detectable by flow cytometry, fluorescence microscopy, or fluorimetry (at excitation/emission spectra of 495 nm/529 nm). Briefly, THP-1 cells cultured in triplicate wells of 12-well plates at a cell density of 10^6^ cells/mL/well were incubated at 37 °C with 10 mM H_2_O_2_ for 10 h. Later, cultured cells were loaded with DCFH-DA probe (15 μM) by incubation at 37 °C for 30 min, and cells were analyzed without washing by flow cytometry.

In the assays involving treatments with other agents, cells cultured in triplicate wells of 12-well plates at a cell density of 10^6^ cells/mL/well were incubated at 37 °C for 24 h with palmitate (200 μM) and/or TNF-α (10 ng/ml) or 0.1% BSA, in the presence or absence of H_2_O_2_ (10mM for 10 h). Cells were harvested for total RNA extraction and supernatants were collected for measuring MIP-1α secreted protein concentrations. In the assays involving antioxidants or ROS inhibitors, THP-1 cells were incubated with *N*-acetyl cysteine (NAC) (5 mM for 40 min), apocynin (100 μM for 1 h), or curcumin (10 μM for 30 min) in designated triplicate wells and then treated with 10 mM H_2_O_2_ for 10 h to induce ROS. Cell supernatants were collected for measuring levels of MIP-1α secreted protein.

Resveratrol targets the TRIF/NF-κB, SIRT1, AMPK, mTOR, and other signaling mediators and is a potent ROS scavenger [30,31]. In assays involving treatments with resveratrol, THP-1 cells cultured in triplicate wells of 12-well plates at a cell density of 10^6^ cells/mL/well were pre-incubated with resveratrol (5 μM) for 30 min or with vehicle only. Later, cells were treated with palmitate (200 μM) and/or TNF-α (10 ng/mL) or 0.1% BSA and incubated at 37 °C for 24 h, in accord with previously published report [15]. Cells were harvested for total RNA extraction and supernatants were collected for measuring concentrations of MIP-1α secreted protein.

### 2.8. Real-Time qRT-PCR

Total RNA was extracted using RNeasy Mini Kit (Qiagen, Hilden, Germany) and following the manufacturer’s instructions. Then, cDNA was synthesized from total RNA (1 μg) and following the given instructions (High Capacity cDNA Reverse Transcription Kit, Applied Biosystems, Foster City, CA, USA). For real-time PCR, cDNA (50 ng) was amplified with Inventoried TaqMan Gene Expression Assay products (MIP-1α: Hs04194942_s1; GAPDH: Hs03929097_g1; TLR4: Hs00152939_mL; and IRF3: Hs01547283_m1) comprising two gene-specific primers, one TaqMan MGB probe (6-FAM dye-labeled), and TaqMan^®^ gene expression master mix (Applied Biosystems) using a 7500 fast real-time PCR system (Applied Biosystems). *MIP-1α* mRNA expression was normalized against GAPDH mRNA expression and target gene expression relative to control was calculated by using 2^−ΔΔCT^ method, expressed as fold change over average gene expression in control treatment taken as 1.

### 2.9. ELISA

MIP-1α-secreted protein levels were measured in the supernatants of THP-1 cells stimulated with palmitate (200 μM) and/or TNF-α (10 ng/mL) or 0.1% BSA using human MIP-1α DuoSet ELISA kit and following the manufacturer’s instructions (Cat. #DY270, R&D Systems Inc.). Briefly, 96-well microplates were prepared by coating overnight with diluted capture antibody (100 μL/well). After three washes, plates were blocked by adding 300 μL of Reagent Diluent to each well and incubating at room temperature for 1 h. After three washes as before, appropriately diluted standards, controls, and samples were added in duplicate wells (100 μL/well), and plates were incubated at room temperature for 2 h. After three washes, Streptavidin-HRP was added (100 μL/well) and incubated at room temperature for 20 min in dark. After washing thrice, substrate solution was added (100 μL/well) and plates were again incubated at room temperature for 20 min in dark. Eventually, 50 μL of stop solution was added to each well and the plates were gently tapped to ensure thorough mixing. The optical density (O.D.) was read using a microplate reader at 450 nm, with wavelength correction set to 540 nm or 570 nm. For measuring MIP-1α concentrations, averages of duplicate readings for each standard, control, and sample were calculated, and the average zero standard O.D. was subtracted from each value. A standard curve was generated by plotting the mean absorbance for each standard on the Y-axis against the concentration on the X-axis to draw a best fit curve through the points on the graph. In order to calculate the final concentrations of MIP-1α (pg/mL), concentrations read from the standard curve were multiplied by the dilution factor as required.

### 2.10. Flow Cytometry

THP-1 cells were cultured (10^6^ cells/mL/well) in triplicate wells of 12-well plates and stimulated with palmitate (200 μM) and/or TNF-α (10 ng/mL) or 0.1% BSA (Sigma) and incubated at 37 °C for 24 h, as previously described. Brefeldin A (eBioscience Cat. #00-4506-51, San Diego, CA, USA) was added (1 μg/mL) to wells during the last 8 h of incubation. Cells were harvested by centrifugation and stained for intracellular expression of MIP-1α following the manufacturer’s instructions. Briefly, cells were washed thrice and then fixed and permeabilized by using fixation/permeabilization buffer (eBioscience Cat. #00-5523-00, San Diego, CA, USA) and incubation for 20 min on wet ice. Cells were washed as before and incubated with mouse anti-human PE-conjugated MIP-1α antibody (BD PharmingenTM Cat. #554730, BD Biosciences, San Jose, CA, USA) for 40 min on wet ice in dark. Again, cells were washed thrice, harvested by centrifugation, and resuspended in flow cytometry buffer. The data (10,000 events) were acquired on BD FACS Canto II flow cytometer (BD Biosciences, San Jose, CA, USA). Mean fluorescence intensity (MFI) and staining index (SI) were determined and analyzed using BD FACSDivaTM Software 8 (BD Biosciences).

### 2.11. Western Blotting

THP-1 cells were harvested and lysed by incubation for 30 min with lysis buffer containing Tris (62.5 mM; pH 7.5), 1% Triton X-100, and 10% glycerol. Supernatants were collected by centrifuging cell lysates at 14,000 rpm for 10 min, and protein concentration was measured by using Quick Start Bradford-13 Dye Reagent and Bio-Rad protein assay kit (Hercules, CA, USA). Samples (20 μg) were mixed with loading buffer, heated at 95 °C for 5 min, and resolved by 12% SDS-PAGE. Fractionated proteins were electroblotted to an Immun-Blot PVDF membrane (Bio-Rad, Hercules, CA, USA) for transfer, blocked for 1h using 5% nonfat milk in PBS, and incubated at 4 °C overnight with primary Abs (1:100 dilution; Cell Signaling Technology Inc., Danvers, MA, USA) against p-C-Jun and p-NF-κB. Blots were washed four times with Tris buffered saline with 0.1% Tween 20 and incubated for 2 h with HRP-conjugated secondary antibody (Promega, Madison, WI, USA). Immunoreactive bands were developed using Amersham ECL Plus Western Blotting Detection System (GE Health Care, Buckinghamshire, UK) and visualized by Molecular Imager® VersaDoc™ MP Imaging Systems (Bio-Rad Laboratories, Irvine, CA, USA).

### 2.12. Statistical Analysis

Data are shown as mean ± SEM values and statistical analysis was performed using Prism 8.3.1 software (GraphPad Inc., San Diego, CA, USA). Group variances were compared by using one-way ANOVA and unpaired Student t-test was used to compare group means. All *P*-values < 0.05 were considered as significant.

## 3. Results

### 3.1. Increased MIP-1α Expression in THP-1 Cells and Primary Human Macrophages Co-Stimulated with Palmitate and TNF-α

The long chain, saturated fatty acid palmitate and the proinflammatory cytokine TNF-α are known to modulate several inflammatory signaling pathways. Elevated levels of these two agents and MIP-1α chemokine are co-expressed in obesity. We asked if palmitate and TNF-α could co-induce MIP-1α expression in THP-1 human monocytic cells and primary human macrophages. To this end, our data show that, relative to vehicle-treated control, *MIP-1α* gene expression was upregulated following treatment of THP-1 monocytic cells with palmitate (22.50 ± 1.33-fold) or TNF-α (25.10 ± 0.40-fold). Interestingly, co-stimulation of cells with palmitate and TNF-α induced a synergistic response with a 295.70 ± 3.78-fold increase in *MIP-1α* gene expression (Figure 1A). As expected, we also detected elevated concentrations of MIP-1α secreted protein in cell supernatants following treatment with palmitate (81.97 ± 0.26 pg/mL) or TNF-α (607.3 ± 0.78 pg/mL), compared to control (33.73 ± 1.85 pg/mL). However, co-stimulation of cells with palmitate and TNF-α led to a substantial increase in the levels of MIP-1α secreted protein in cell supernatants (1406.00 ± 5.23 pg/mL) (Figure 1B). Notably, similar data were obtained using primary human macrophages isolated from blood samples of three healthy donors. In this regard, solitary treatments of primary macrophages with palmitate and TNF-α-induced upregulated *MIP-1α* gene expression of 48.50 ± 7.50-fold and 119.00 ± 9.00-fold, respectively, compared to control. However, *MIP-1α* gene expression was upregulated by 182.50 ± 3.50-fold as macrophages were co-stimulated with palmitate and TNF-α (Figure 1C). Consistent with the changes in *MIP-1α* gene expression, elevated levels of MIP-1α secreted protein were also detected in supernatants as macrophage cultures were stimulated with palmitate (736.70 ± 0.94 pg/mL) or TNF-α (799.60 ± 1.10 pg/mL) compared to control (132.80 ± 17.33 pg/mL). Again, higher levels of MIP-1α secreted protein (1107.00 ± 21.01 pg/mL) were detected in supernatants as macrophages were co-stimulated with palmitate and TNF-α, compared to control (Figure 1D). In addition, a similar pattern of upregulation in MIP-1α secreted protein was observed as PBMC were co-stimulated with palmitate and TNF-α (245.70 ± 1.21 pg/mL) as compared to stimulation with either palmitate (69.40 ± 1.52 pg/mL) or TNF-α (148.50 ± 0.41 pg/mL) (Appendix A). The intracellular expression of MIP-1α protein in THP-1 monocytic cells was also confirmed by flow cytometry. As expected, intracellular MIP-1α protein expression was remarkably higher in cells that were co-stimulated with palmitate and TNF-α (Staining Index “SI” = 148.56) compared to those stimulated with either palmitate (SI = 93.52) or TNF-α (SI = 85.39) (Figure 1E–G). MIP-1α protein expression co-induced by palmitate and TNF-α differed significantly from that induced by either palmitate or TNF-α alone (*P* = 0.003; Figure 1H). Taken together, MIP-1α expression is upregulated at the transcriptional and translational levels as THP-1 monocytic cells or primary macrophages are stimulated with either palmitate or TNF-α; however, co-stimulation with palmitate and TNF-α further drives the MIP-1α expression, indicating a cooperative effect between two agonistic stimuli.

### 3.2. MIP-1α Co-Induction by Palmitate and TNF-α Involves the TLR4-Mediated Signaling and Clathrin-Mediated Endocytosis (CME)

TLR4 activation can induce expression of MIP-1α and other chemokines in both mice and humans [21,32]. Given that, we speculated that TLR4 genetic ablation could lead to suppression of MIP-1α following stimulation with palmitate and TNF-α. To this end, as expected, co-stimulation of TLR4 siRNA-transfected monocytic cells with palmitate and TNF-α led to suppression of *MIP-1α* transcripts (mean decrease from 20.30-fold to 12.25-fold; *P* = 0.02), and MIP-1α secreted protein (mean decrease from 63.01 pg/mL to 15.59 pg/mL; *P* = 0.0002) as compared to respective controls transfected with scrambled siRNA (Figure 2A,B). These changes were concordant with a significant ablation of TLR4 (*P* = 0.02) in cells that were transfected with TLR4-specific siRNA (Appendix A). Next, we resorted to TLR4 receptor labeling with anti-TLR4 neutralizing antibody before the cells were co-stimulating with palmitate and TNF-α. Interestingly, our previous observations were recapitulated as the antibody-labeled cells displayed a drastic suppression of *MIP-1α* transcripts (mean decrease from 296.27 fold to 43.57 fold; *P* = 0.002) and MIP-1α secreted protein (mean decrease from 1359.85 pg/mL to 816.88 pg/mL; *P* = 0.01) compared to respective controls labeled with isotype-matched antibody instead (Figure 2C,D). These data were further validated as THP-1 cells were treated with a TLR4 inhibitor OxPAPC before co-stimulation with palmitate and TNF-α. As expected, OxPAPC-treated cells had a lower expression of *MIP-1α* transcripts (mean decrease from 741.20-fold to 294.70-fold; *P* = 0.0007) and MIP-1α secreted protein (mean decrease from 128.60 pg/mL to 65.15 pg/mL; *P* = 0.003) as compared to respective controls treated with vehicle only (Figure 2E,F).

Next, we investigated the role of non-canonical TRIF-dependent signaling associated with palmitate internalization via the CME. To this effect, THP-1 cells were treated with CME inhibitor CPZ before co-stimulation with palmitate and TNF-α. To this end, our data show that blocking CME reduced the cooperative induction of MIP-1α by palmitate and TNF-α as a lower expression of *MIP-1α* mRNA (mean decrease from 295.66-fold to 177.35-fold; *P* = 0.004) and MIP-1α secreted protein (mean decrease from 1406.41 pg/mL to 1135.91 pg/mL; *P* = 0.008) was observed in CPZ-treated cells as compared to vehicle-treated controls (Figure 3).

### 3.3. MIP-1α Co-Induced by Palmitate and TNF-α Involves the IRF3 Pathway

TLR4 downstream signaling following ligand binding is mediated by MyD88-dependent and MyD88-independent (TRIF/IRF3) pathways [33]. We asked whether the MIP-1α expression following co-stimulation with palmitate and TNF-α required MyD88- or IRF3-mediated signaling. To test the involvement of MyD88 adaptor protein, MyD88-null or THP1-XBlue™-defMyD cells were used and the lack of MyD88 did not abolish MIP-1α expression as cells were co-stimulated with palmitate and TNF-α (Figure 4A,B). In line with our previous data using MyD88-competent cells, increased levels of *MIP-1α* transcripts (mean increase from 92.16-fold to 919.90-fold; *P* < 0.0001) and MIP-1α secreted protein (mean increase from 416.40 pg/mL to 669.20 pg/mL; *P* < 0.0001) were observed using MyD88-null cells that were co-stimulated with palmitate and TNF-α as compared to controls treated with either palmitate or TNF-α, implicating that co-induction of MIP-1α by palmitate and TNF-α most likely involved the MyD88-independent (TRIF/IRF3) pathway. To test this, IRF3 was genetically ablated by transfecting THP-1 cells with IRF3-specific siRNA, which resulted in a significant suppression of IRF3 mRNA (Appendix A). As expected, co-stimulation of IRF3-ablated cells with palmitate and TNF-α led to a diminished expression of *MIP-1α* transcripts (mean decrease from 20.90-fold to 15.40-fold; *P* = 0.006) and MIP-1α secreted protein (mean decrease from 727.70 pg/mL to 277.70 pg/mL; *P* = 0.001) compared to scrambled siRNA controls (Figure 4C,D). Taken together, these data suggest that MIP-1α co-induced by palmitate and TNF-α in THP-1 cells involves the IRF3 pathway.

In addition to determining the effect of palmitate, which is a non-canonical TLR4 activator, we also tested for comparison the effect of a classical TLR4 agonist LPS on MIP-1α protein expression and the effect of palmitate on a classical TLR4-related cytokine IL-6. The representative data from three independent experiments with similar results showing the induction of MIP-1α mRNA and protein in THP-1 cells by palmitate, LPS, TNF-α, palmitate+TNF-α, and LPS+TNF-α are presented in Appendix A. Similarly, the representative data from three independent experiments showing the induction of IL-6 mRNA and protein in THP-1 cells by stimulation with palmitate, LPS, TNF-α, palmitate+TNF-α, and LPS+TNF-α are presented in Appendix A. These data collectively indicate that a canonical TLR4 activation by LPS resulting in MIP-1α expression is more potent than a non-canonical TLR4 activation by palmitate. As well, palmitate is less potent in inducing a classical TLR4 related cytokine IL-6 than the levels induced by LPS.

### 3.4. MIP-1α Co-Induced by Palmitate and TNF-α Involves Signaling Via c-Jun and NF-κB

To induce/promote inflammatory gene expression in monocytes/macrophages and other cells, TNF-α and palmitate activate the signal transduction by phosphorylating NF-κB and/or c-Jun *N*-terminal kinases (JNKs) [34]. To determine the involvement of c-Jun and NF-κB mediated signaling, THP-1 monocytic cells were stimulated with palmitate (200 μM for 1 h) and/or TNF-α (10 ng/mL for 10 min) and cell lysates were analyzed for expression of c-Jun and NF-κB phosphorylated and total proteins. As shown by western blotting (Figure 5A), co-stimulation of THP-1 cells with palmitate and TNF-α induced stronger phosphorylation of c-Jun (*P* = 0.03) and NF-κB (*P* = 0.002) as compared to individual treatments with palmitate or TNF-α (Figure 5B,C, respectively).

### 3.5. MIP-1α Expression Induced by Palmitate and/or TNF-α, in Presence or Absence of Oxidative Stress

Treatment of monocytes/macrophages and other cells with palmitate and/or TNF-α may induce oxidative stress in cells to various levels, depending on the type of cell, stimulus and length of exposure [35]. We wanted to compare ROS induction in THP-1 monocytic cells following treatments with a known ROS-inducer as well as with palmitate and/or TNF-α. To this effect, we induced ROS induction by treating THP-1 cells with H_2_O_2_ using the concentrations and protocol already optimized in our laboratory. In parallel, we also treated cells with palmitate and/or TNF-α and intracellular ROS expression was measured by DCFH-DA assay using flow cytometry. As expected, the data show induction of ROS following cell treatments with H_2_O_2_ (SI = 23.10), palmitate (SI = 5.89), TNF-α (SI = 2.73) and palmitate+TNF-α (SI = 33.86) (Figure 6A–D, respectively). The ROS induction by H_2_O_2_ was higher than either palmitate or TNF-α alone (*P* = 0.002); however, it was lower than that induced by co-stimulation with palmitate and TNF-α (*P* = 0.008) (Figure 6E). Next, cells were treated with palmitate and/or TNF-α, in presence or absence of H_2_O_2_, and *MIP-1α* gene expression was determined. *MIP-1α* mRNA expression was upregulated in cells that were treated with H_2_O_2_ alone (mean increase from 1.00 fold to 26.50 fold; *P* = 0.0002), H_2_O_2_ plus palmitate (mean increase from 4.50 fold to 43.00-fold; *P* < 0.0001), and H_2_O_2_ plus TNF-α (mean increase from 8.50-fold to 46.50-fold; *P* < 0.0001) compared to respective controls without H_2_O_2_. Further, levels of *MIP-1α* mRNA co-induced by palmitate and TNF-α, with H_2_O_2_ (60.67 ± 2.03-fold) or without H_2_O_2_ (63.00 ± 6.00-fold) differed non-significantly (*P* = 0.98) (Figure 6F). In agreement with gene expression, MIP-1α protein secretion was also amplified as THP-1 cells were treated with H_2_O_2_ alone (mean increase from 5.50 pg/mL to 292.80 pg/mL; *P* < 0.0001), H_2_O_2_ plus palmitate (mean increase from 107.60 pg/mL to 284.50 pg/mL; *P* < 0.0001), H_2_O_2_ plus TNF-α (mean increase from 306.90 pg/mL to 512.80 pg/mL; *P* < 0.0001), and H_2_O_2_ plus palmitate and TNF-α (mean increase from 700.70 pg/mL to 745.20 pg/mL; *P* = 0.002) (Figure 6G).

### 3.6. MIP-1α Induction by Oxidative Stress Is Counteracted by ROS/NF-κB Inhibitors

We next asked whether the ROS-driven MIP-1α expression in monocytic cells could be reduced by treatments with ROS and/or NF-κB inhibitors. To this end, we found that H_2_O_2_-induced MIP-1α protein expression in THP-1 cells was significantly reduced after treatment with NAC (302.20 ± 82.76 pg/mL; *P* = 0.002), apocynin (360.90 ± 107.50 pg/mL; *P* = 0.01), and curcumin (209.90 ± 46.27 pg/mL; *P* = 0.003) compared to controls treated with H_2_O_2_ only (682.30 ± 10.60 pg/mL) (Figure 7A).

To determine the effects of an inhibitor targeting both ROS and NF-κB, THP-1 cells were incubated with a stilbenoid called resveratrol (3,5,4′-trihydroxy-*trans*-stilbene) [36]. As expected, diminished expression of *MIP-1α* mRNA (mean decrease from 295.70-fold to 156.11-fold; *P* = 0.01) and MIP-1α secreted protein (mean decrease from 1406.80 pg/mL to 1169.00 pg/mL; *P* = 0.002) was observed in resveratrol-treated cells compared to vehicle-treated controls as both types of cells were co-stimulated with palmitate and TNF-α (Figure 7B,C, respectively).

Overall, our data suggest a “cooperativity model of MIP-1α amplification” in monocytic cells that are subjected to co-stimulation with palmitate, TNF-α, and ROS-mediated oxidative stress (Figure 8).

## 4. Discussion

Given that palmitate, TNF-α, and MIP-1α have emerged as critical players in metabolic inflammation, their elevated expression in obesity remains a significant concern, and it is unclear whether a cooperative interaction between palmitate and TNF-α can lead to upregulate MIP-1α levels in human monocytic cells and/or macrophages. In this study, we report two novel findings, to our knowledge, as follows. First, palmitate and TNF-α interact cooperatively to enhance the expression of MIP-1α in THP-1 monocytic cells or macrophages through the TLR4-IRF3 dependent mechanism. Second, ROS-mediated oxidative stress further amplifies this MIP-1α expression.

Palmitate activates inflammatory pathways in primary microvascular endothelial cells, impairs insulin signaling, and enhances monocytic transmigration [37]. TNF-α is also known to contribute to metabolic inflammation and insulin resistance, while in obesity, increased levels of TNF-α have been detected in adipose tissue and skeletal muscle together with infiltration of activated monocytes/macrophages [38]. Our data show that palmitate and TNF-α cooperativity induces MIP-1α expression in monocytic cells/macrophages which is consistent, at least in part, with findings of a previous study on CCL2 expression [15]. TLRs are emerging as nutrient sensors and palmitate and LPS act as TLR4 agonists [13,39]. Herein, we also show that the cooperativity between palmitate and TNF-α leading to MIP-1α expression was hinged on signaling through the TLR4 receptor. In this regard, we demonstrate that siRNA-mediated genetic ablation of TLR4 in human monocytic cells diminishes the expression of *MIP-1α* transcripts as well as MIP-1α secreted protein. We also show that TLR4 blocking by neutralizing antibody reduces the MIP-1α expression at both transcriptional and translational levels. Chemical inhibition of TLR4-mediated signaling by OxPAPC also obviates the co-induction of MIP-1α by palmitate and TNF-α. Thus, several lines of evidence lead to the finding that this cooperative induction of MIP-1α involves the TLR4-mediated signaling. This is in agreement, at least in part, with other reports indicating the involvement of TLR4 in palmitate-induced cytokine/chemokine expression in monocytic cells [21,40].

Proteins, metabolites, and hormones traffic across the cell membrane through assembly and maturation of clathrin-coated pits that gather the cargo as they invaginate and pinch off to form clathrin-coated vesicles during the CME. It is a major endocytic pathway in mammalian cells and a key regulatory mechanism for internalization and recycling of receptors that are engaged in nutrient uptake, signal transduction and synaptic vesicle reformation [41]. In this regard, our data show that MIP-1α expression co-induced by palmitate and TNF-α involves the CME pathway for TLR4 endocytic trafficking as MIP-1α expression was diminished in monocytic cells that were treated with chlorpromazine. This is corroborated by other studies showing that CME pathway is critical to TLR4-mediated signaling [42,43].

Of note, TLR4-downstream signaling may or may not involve MyD88 as an adaptor protein [44]. We next investigated whether or not the cooperative induction of MIP-1α by palmitate and TNF-α required MyD88. To this end, our data show that MIP-1α gene and protein expression was not abolished in MyD88-defective THP1-XBlue defMyD cells. On the contrary, MIP-1α transcripts and secreted protein were significantly diminished in IRF3-ablated cells following co-stimulation with palmitate and TNF-α. This implies that the cooperative expression of MIP-1α was IRF3-dependent; which is contrary to MyD88-dependent expression of MIP-1β in THP-1 cells [40]. The involvement of NF-κB/MAPK signaling pathways was indicated by hyperphosphorylation NF-κB and c-Jun signaling proteins as detected by western blotting. In agreement to this, TLR4 activation and downstream signaling is known to involve the NF-κB and MAPK pathways [45]. Similarly, TLR4 and/or TNF-αR agonists have been shown to activate the NF-κB/AP-1 mediated signaling [46]. In concordance with these reports, NF-κB/MAPK dependent expression of inflammatory cytokines/chemokines was demonstrated in monocytes, T-lymphocytes, alveolar macrophages, plasmacytoid dendritic cells, microglial cells, and chondrosarcoma cells [21,47].

ROS in living cells are normally produced during the activity of various cellular systems that are localized in the (1) plasma membrane (membrane-bound NADPH oxidases, cyclooxygenases, lipoxygenases); (2) cytosol (xanthine oxidase); (3) mitochondria (oxidative phosphorylation for ATP synthesis, by super oxide dismutase and oxidoreductases); (4) endoplasmic reticulum (microsomal cytochrome P450-dependent system, NADH cytochrome b5 reductase and desaturases, protein disulfide isomerase); (5) peroxisomes (fatty acid β- and α-oxidation, iNOS, peroxisomal and xanthine oxidases); and (6) lysosomes (acidic hydrolases). The cellular ROS generated by these diverse sources are tightly regulated by homeostatic mechanisms involving anti-oxidative enzymes, and this homeostasis is impaired under conditions of cellular stress, senescence, and disease. Oxidative stress by ROS plays as an early trigger of metabolic inflammation and related pathobiological changes. We, therefore, asked if the ROS-mediated oxidative stress could amplify MIP-1α levels after palmitate/TNF-α co-stimulation. Our data show that H_2_O_2_ treatment of monocytic cells upregulates the intracellular ROS expression and also amplifies the MIP-1α secreted protein in response to treatments with palmitate, TNF-α, or both. Mechanistically, ROS changes the redox state of nuclear proteins as well as activates transcriptional factors including c-Fos, c-Jun, and NF-κB/Rel [48]. Thus, H_2_O_2_ upregulates the expression of MIP-1α in human monocytic cells by acting as a second messenger. Similar changes in other chemokines including MCP-1, MIP-1β, and CXCL2 have been reported in murine macrophages [49]. In our study, ROS-mediated oxidative stress led to increased levels of *MIP-1α* mRNA and protein in THP-1 cells after treatment with palmitate or TNF-α. Nonetheless, MIP-1α protein, but not mRNA, was upregulated as cells were co-stimulated with H_2_O_2_, palmitate, and TNF-α. Regarding this discrepancy between MIP-1α protein and gene expression, we speculate that the changes in stability of mRNA transcripts and protein turnover rates may play a role in the regulation of proinflammatory genes. Given that, the changes in the turnover rates can influence the steady-state levels of mRNAs over time. In line with this argument, cellular growth rates were shown to impact the overall mRNA turnover which modulated the transcription or degradation rates of specific gene regulons [50]. In agreement with our data showing a ROS-driven MIP-1α amplification, MIP-2 expression was reported to be enhanced in a mouse model of oxidant stress [51].

Furthermore, we performed the inhibitory studies to test whether the ROS-induced MIP-1α expression in monocytic cells could be prevented by using ROS scavengers or antioxidants. To this effect, our data show a significant reduction in the MIP-1α expression after treatments with NAC, apocynin, and curcumin. NAC, a *N*-acetyl derivative of L-cysteine and a precursor of the reduced glutathione (GSH), acts as an antioxidant [52]. Apocynin is a NADPH oxidase (NOX) inhibitor and acts by scavenging the superoxide anion (O_2_^−•^), which is generated during the reduction of O_2_ by NOX in order to abate the oxidative stress and inflammation [53]. Curcumin, also known as diferuloylmethane (1,7-bis(4-hydroxy-3-methoxyphenyl)-1,6-heptadiene-3,5-dione), is a natural polyphenol found in the rhizome of *Curcuma longa* (turmeric) and is known to have antioxidant, antiinflammatory, antimicrobial, and antimutagenic or anticancer properties [54]. We further found that resveratrol, an inhibitor of several targets such as ROS, NF-κB, and mTOR, reduced the expression of MIP-1α at gene and protein levels. Our data indicating the suppression of MIP-1α with these agents are corroborated by other reports that also show the reduced inflammatory cytokines/chemokines and inflammation in relation to these substances [55,56]. Herein, we also show that a canonical TLR4 agonist LPS is more potent in inducing MIP-1α than a non-canonical inducer palmitate, whether alone or in combination with TNF-α. Overall, these observations underscore the role of ROS (oxidative stress) in metabolic inflammation through the induction of MIP-1α expression in THP-1 monocytic cells. However, our study involves minor caveats such as additional data regarding adipose MIP-1α expression in obesity was not addressed as well as monocytes/macrophage or neutrophil chemotaxis assay of MIP-1α was not performed. The future studies will include in vivo investigations to test whether the interventions including antioxidants or ROS scavengers can prevent or mitigate the inflammatory responses in metabolic disease setting such as obesity or type-2 diabetes.

In conclusion, our data show that palmitate and TNF-α co-induce MIP-1α expression in THP-1 monocytic cells or primary human macrophages. The underlying molecular mechanisms involve TLR4-IRF3/CME pathways and signaling via c-Jun/NFκB. Oxidative stress via ROS elevates the MIP-1α expression. Together, these findings support a “cooperativity model of MIP-1α amplification” hinged on co-stimulation with palmitate, TNF-α, and oxidative stress, which may have significance for metabolic inflammation.

## Figures and Tables

**Figure 1 cells-09-01799-f001:**
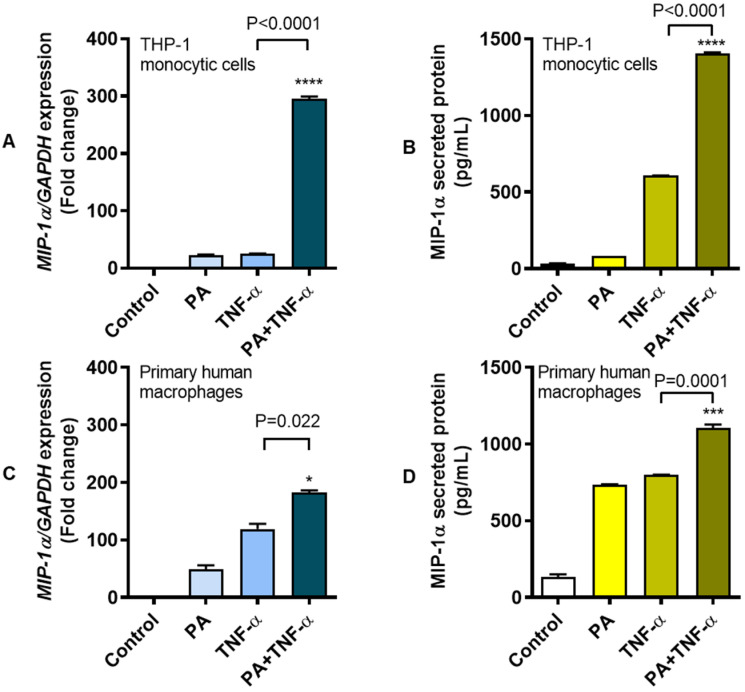
Enhanced expression of MIP-1α in THP-1 monocytic cells and primary human macrophages co-stimulated with palmitate and TNF-α. Human monocytic THP-1 cells and primary human macrophages were stimulated with palmitate (200 μM) and/or TNF-α (10 ng/mL), while control wells were treated with 1%BSA. Cells were incubated at 37 °C for 24 h and cells were collected for total RNA extraction for measuring MIP-1α gene expression and supernatants were collected for measuring MIP-1α secreted protein as described in the Materials and Methods section. The representative data (mean ± SEM) from 3 independent determinations with similar results show significantly increased expression of (**A**) *MIP-1α* mRNA (*P* < 0.0001) and (**B**) MIP-1α secreted protein (*P* < 0.0001) in THP-1 monocytic cells co-stimulated with palmitate and TNF-α compared to stimulation with either TNF-α or palmitate. Similarly, the representative data (mean ± SEM) from three independent determinations with similar results show significantly elevated expression of (**C**) *MIP-1α* transcripts (*P* = 0.022) and (**D**) MIP-1α secreted protein (*P* = 0.0001) in primary human macrophages co-stimulated with palmitate and TNF-α compared to stimulation with either TNF-α or palmitate. (**E**–**G**) The representative flow cytometry histograms from 3 independent experiments with similar results also show the upregulated intracellular protein expression of MIP-1α in THP-1 cells co-stimulated with palmitate and TNF-α (SI (staining index) = 148.56) compared to those stimulated with either palmitate (SI = 93.52) or TNF-α (SI = 85.39). (**H**) MIP-1α protein expression induced by co-stimulation with palmitate and TNF-α was significantly higher than that induced by palmitate or TNF-α alone (*P* = 0.003). SI: Staining index; * *P* < 0.05, ** *P* < 0.01, *** *P* < 0.001 and **** *P* < 0.0001.

**Figure 2 cells-09-01799-f002:**
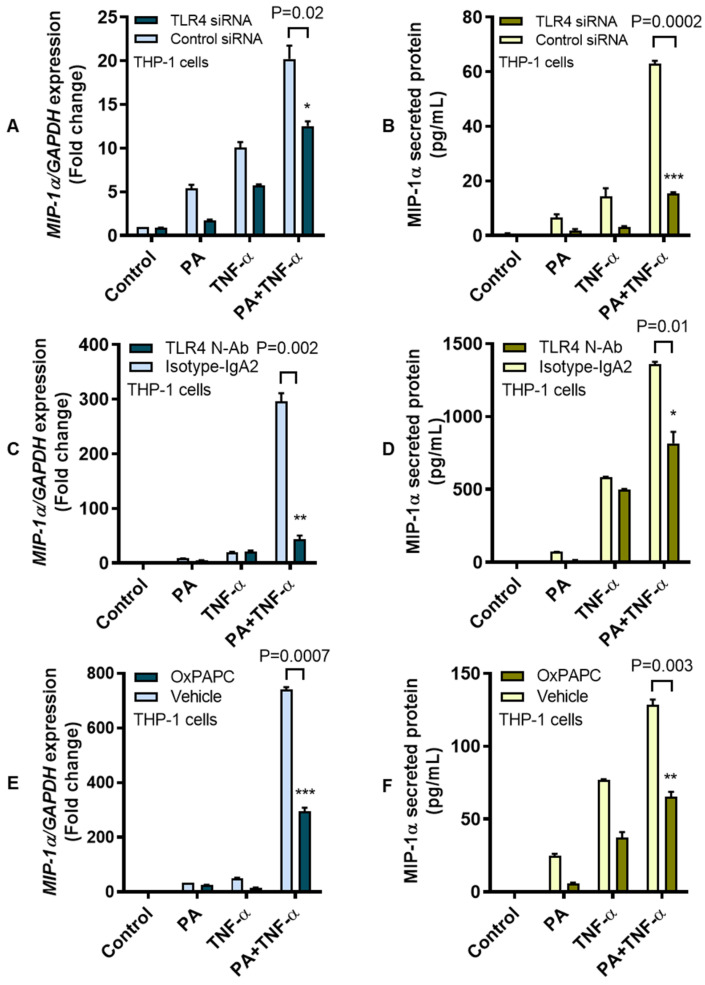
MIP-1α co-induction by palmitate and TNF-α implicates the TLR4-mediated signaling. TLR4 signaling activity in THP-1 cells was intercepted by multiple approaches such as genetic ablation with TLR4-specific siRNA, receptor blocking with TLR4 neutralizing antibody, and chemical inhibition with TLR4 inhibitor OxPAPC as described in the Materials and Methods section. The representative data (mean ± SEM) from three independent determinations with similar results show that co-induction of MIP-1α by palmitate and TNF-α was suppressed at the levels of (**A**) *MIP-1α* transcripts (*P* = 0.02) and (**B**) MIP-1α secreted protein (*P* = 0.0002) in THP-1 monocytic cells transfected with TLR4 siRNA as compared to controls transfected with scrambled siRNA. Similarly, MIP-1α expression co-induced by palmitate and TNF-α was reduced at the levels of (**C**) *MIP-1α* mRNA (*P* = 0.002) and (**D**) MIP-1α secreted protein (*P* = 0.01) in THP-1 cells that were treated with anti-TLR4 neutralizing antibody compared to controls treated with isotype-matched IgA2 antibody. In line with these results, treatment of THP-1 monocytic cells with OxPAPC inhibitor also reduced the expression of (**E**) *MIP-1α* transcripts (*P* = 0.0007) and (**F**) MIP-1α secreted protein (*P* = 0.003) compared to controls treated with the vehicle only before co-stimulation with palmitate and TNF-α. * *P* < 0.05, ** *P* < 0.01, *** *P* < 0.001.

**Figure 3 cells-09-01799-f003:**
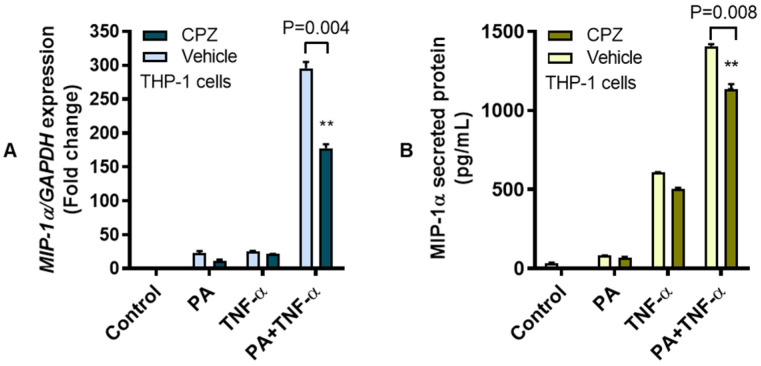
Co-induction of MIP-1α by palmitate and TNF-α involves clathrin-mediated endocytosis. To determine whether the cooperative induction of MIP-1α involved clathrin-mediated endocytosis (CME), THP-1 monocytic cells were treated with the CME inhibitor chlorpromazine CPZ (20 μM for 1 h) before stimulation with palmitate (200 μM) and/or TNF-α (10 ng/mL), while control wells were treated with 1%BSA. Cells were incubated at 37 °C for 24 h and cells were collected for total RNA extraction for measuring MIP-1α gene expression and supernatants were collected for measuring MIP-1α secreted protein as described in the Materials and Methods section. The representative data (mean ± SEM) obtained from three independent determinations with similar results show that CPZ-treated cells displayed the reduced expression of (**A**) *MIP-1α* mRNA (*P* = 0.004) and (**B**) MIP-1α secreted protein (*P* = 0.008) compared to controls treated with the vehicle only before co-stimulation with palmitate and TNF-α. ** denotes a *P*-value < 0.01.

**Figure 4 cells-09-01799-f004:**
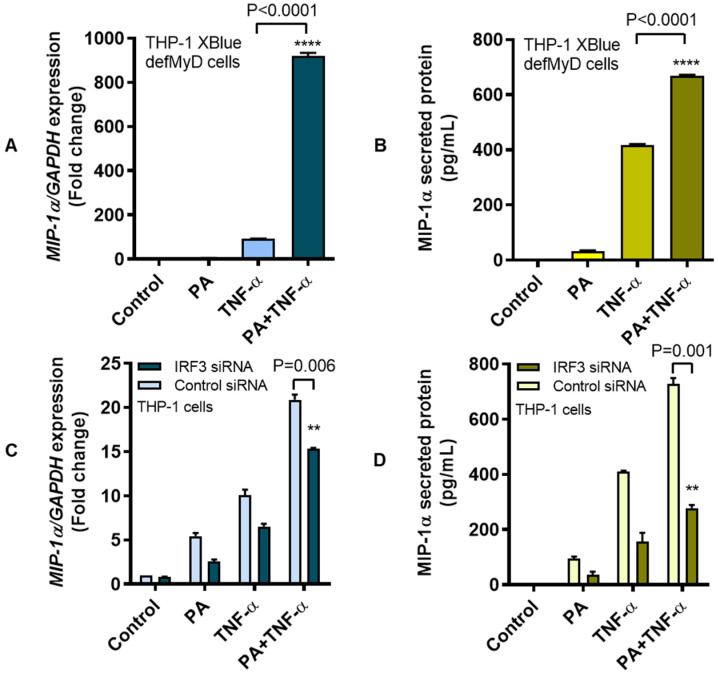
MIP-1α co-induction by palmitate and TNF-α involves the IRF3 pathway. To investigate whether the MIP-1α co-induction by palmitate and TNF-α required MyD88 adaptor protein for TLR4-downstream signaling, we used MyD88-null THP1-XBlue-defMyD cells and treated cell cultures at 37 °C with palmitate (200 μM) and/or TNF-α (10 ng/mL) for 24 h. Cell pellets were collected for total RNA extraction for measuring MIP-1α gene expression and cells supernatants were collected for measuring MIP-1α secreted protein as described in the Materials and Methods section. The representative data (mean ± SEM) from three independent determinations with similar results show that MIP-1α co-induction by palmitate and TNF-α was sustained in MyD88-defective cells which showed elevated expression of (**A**) *MIP-1α* transcripts (*P* < 0.0001) and (**B**) MIP-1α secreted protein (*P* < 0.0001) in response to co-stimulation with palmitate and TNF-α as compared to those treated with either TNF-α or palmitate. On the contrary, siRNA-mediated ablation of IRF3 resulted in the diminution of (**C**) *MIP-1α* mRNA (*P* = 0.006) and (**D**) MIP-1α secreted protein (*P* = 0.001) compared to controls transfected with scrambled siRNA. Thus, MIP-1α co-induction in THP-1 cells by palmitate and TNF-α involves the IRF3-mediated signaling. ** *P* < 0.01 and **** *P* < 0.0001.

**Figure 5 cells-09-01799-f005:**
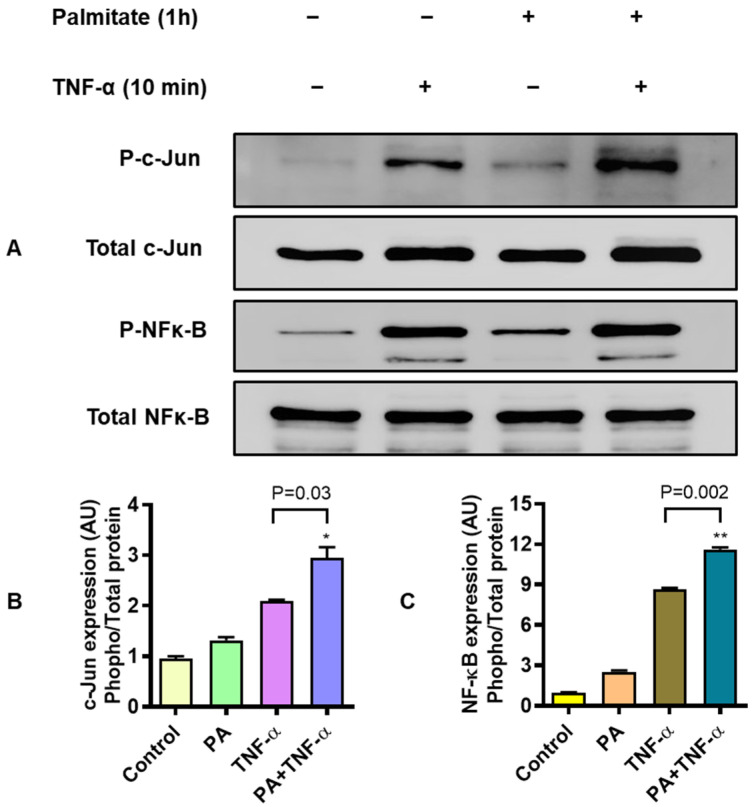
MIP-1α co-induction by palmitate and TNF-α implicates signaling via c-Jun/NF-κB. To test whether the MIP-1α co-induction by palmitate and TNF-α involved c-Jun and NF-κB signaling, THP-1 monocytic cells were treated with palmitate (200 μM for 1 h) and TNF-α (10 ng/mL for 10 min) while control wells were treated with vehicle only. Cells were lysed in RIPA buffer and 20 μg protein from each sample was loaded for analysis by SDS-PAGE. The representative data (mean ± SEM) from three independent determinations with similar results show the downmodulated expression of (**A**) *MIP-1α* transcripts (*P* = 0.01) and (**B**) MIP-1α secretory protein (*P* = 0.002) in resveratrol-treated cells as compared to controls treated with the vehicle only before co-stimulation with palmitate and TNF-α. (**A**) Western blots show the expression of hyperphosphorylated c-Jun (p-cJun) and NF-κB (p-NF-κB) proteins against respective total proteins in THP-1 monocytic cells co-stimulated with palmitate and TNF-α as compared to their expression in cells treated with either palmitate or TNF-α. Normalization of phosphorylated against total proteins shows significantly increased expression of (**B**) phosphorylated c-Jun and (**C**) phosphorylated NF-κB in cells co-stimulated with palmitate and TNF-α, compared to individual stimulations with palmitate or TNF-α. * *P* < 0.05 and ** *P* < 0.01.

**Figure 6 cells-09-01799-f006:**
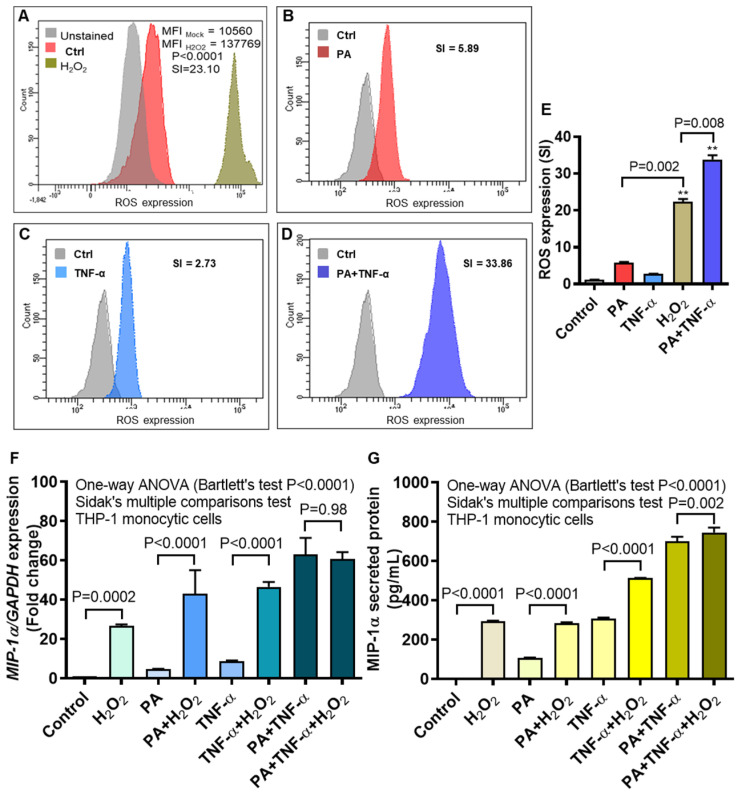
MIP-1α expression by palmitate and/or TNF-α, in presence or absence of oxidative stress. To test whether the oxidative stress was induced by palmitate and/or TNF-α, THP-1 cells were treated with H_2_O_2_ (10 mM for 10 h), palmitate, TNF-α, as well as palmitate+TNF-α and the intracellular expression of reactive oxygen species (ROS) was measured by DCFH-DA assay as described in Section Materials and Methods. The representative flow cytometry data obtained from three independent determinations with similar results show the increased ROS expression in THP-1 cells treated with (**A**) H_2_O_2_ (SI = 23.10), (**B**) palmitate (SI = 5.89), (**C**) TNF-α (SI = 2.73), and (**D**) palmitate+TNF-α (SI = 33.86), indicating induction of the oxidative stress by all treatments. (**E**) ROS induction by H_2_O_2_ was higher (*P* = 0.002) than that induced by palmitate or TNF-α, but lower (*P* = 0.008) than that co-induced by palmitate and TNF-α. Next, THP-1 cells were treated with vehicle (control); palmitate; TNF-α; and palmitate+TNF-α, in the presence or absence of H_2_O_2_. The representative data (mean ± SEM) obtained from three independent determinations with similar results show the upregulated expression of (**F**) *MIP-1α* transcripts and (**G**) MIP-1α secreted protein in the presence of H_2_O_2_ (All *P* < 0.05); except the *MIP-1α* gene expression co-induced by palmitate and TNF-α, with or without H_2_O_2_ (*P* = 0.98). MFI: mean fluorescence intensity; SI: staining index. ** denotes *P*-value < 0.01.

**Figure 7 cells-09-01799-f007:**
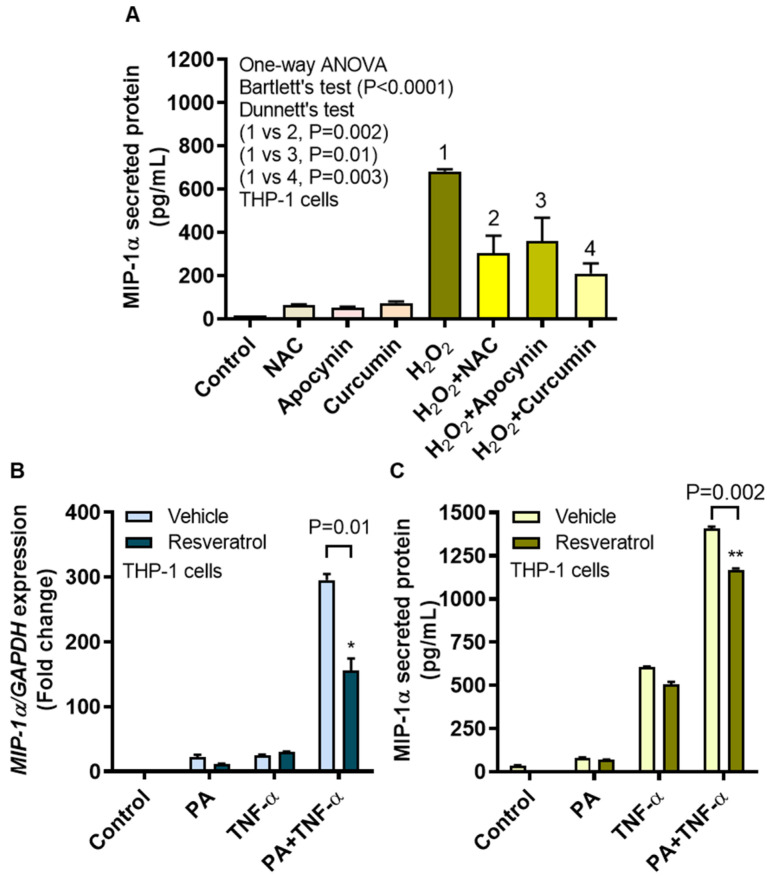
MIP-1α induction by oxidative stress is mitigated by ROS/NF-kB inhibitors. To test whether the MIP-1α induction by oxidative stress was reduced by ROS inhibitors, THP-1 cells were treated with *N*-acetyl cysteine (NAC), apocynin, or curcumin before stimulation with H_2_O_2_ (10 mM for 10 h) and MIP-1α secreted protein expression was measured as described in Section Materials and Methods. To assess the effects of ROS/NF-κB inhibitor on MIP-1α gene and protein expression, cells were pre-treated with resveratrol (5 μM for 30 min) before stimulation with palmitate and/or TNF-α The representative data (mean ± SEM) from three independent determinations with similar results show a significant reduction in (**A**) MIP-1α secreted protein in cells that were treated with NAC (*P* = 0.002), apocynin (*P* = 0.01), or curcumin (*P* = 0.003).The data also show significant reduction in (**B**) MIP-1α mRNA (*P* = 0.01) and (**C**) MIP-1α protein (*P* = 0.002) expression following resveratrol treatment. * *P* < 0.05 and ** *P* < 0.01.

**Figure 8 cells-09-01799-f008:**
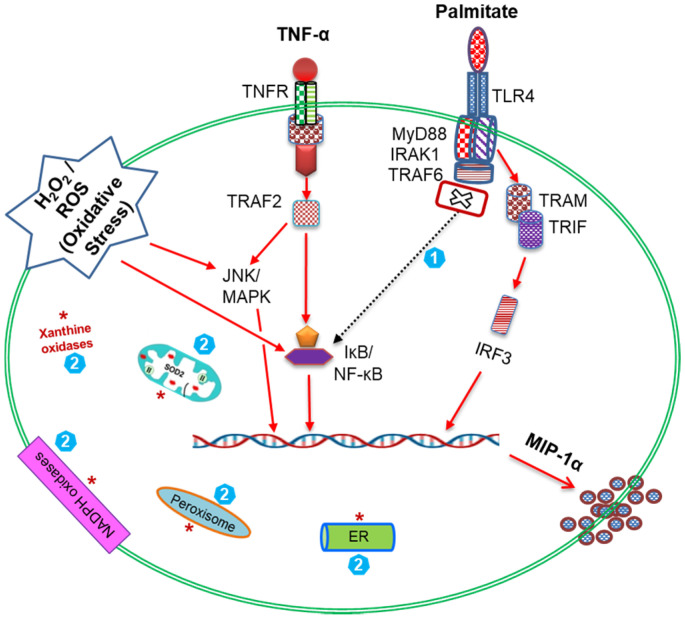
The “cooperativity model of MIP-1α amplification” in THP-1 monocytic cells. A schematic illustration shows that palmitate-TLR4 interaction activates downstream signaling via the TRIF/IRF3 pathway for MIP-1α expression. While, TNF-α-TNFR1 interaction and TRAF2 engagement activate signaling via the NF-κB and JNK/MAPK pathways and lead to MIP-1α expression. Notably, ROS-mediated oxidative stress also leads to activation of the NF-κB and JNK/MAPK pathways and MIP-1α expression. Taken together, the simultaneous exposure of THP-1 cells to the three agonistic stimuli i.e., palmitate, TNF-α and H_2_O_2_ likely transduces a strong cooperative signaling cascade (red arrows) resulting in the amplified MIP-1α expression. Note: (1) This model highlights the novel finding that cooperative MIP-1α expression, unlike MIP-1β expression, does not involve the MyD88 signaling pathway (cross sign with dashed arrow); (2) The intracellular sources of endogenous ROS are shown by red asterisks that include the membrane bound NADPH oxidases, cytoplasmic xanthine oxidases, endoplasmic reticulum, mitochondria, and peroxisomes.

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
