# Peer review of "MIP-1α Expression Induced by Co-Stimulation of Human Monocytic Cells with Palmitate and TNF-α Involves the TLR4-IRF3 Pathway and Is Amplified by Oxidative Stress"

_cells, 2020, doi:10.3390/cells9081799_

Round 1
Reviewer 1 Report
The authors described that enhanced expression of MIP-1α in human monocytic cells co-treated with TNF-α and palmitate through TLR4/IRF3, and c-Jun/NF κB signaling pathway. Overall, the manuscript is well-written. Some concerns described below.
- In Fig.5, Western blotting showed high level of p-NF-κB in TNF-α-treated cells. Why the expression of MIP-1 α protein decreased scarcely in TNF-α-treated cells with resveratrol?
- In Fig. 6, the effect of 10 mM H2O2 (high dosage) on MIP-1α induction in TNF-α and palmitate co-treated cells was not obvious. In my opinion, it was inappropriate to state that “ Oxidative stress further promotes the MIP-1α expression co-induced by palmitate and TNF-α “ . In addition, the role of ROS in the synergistic induction of MIP-1α in the combined treatment needed to be discussed.
Author Response
Response to Reviewer 1 Comments
Comments and Suggestions for Authors
The authors described that enhanced expression of MIP-1α in human monocytic cells co-treated with TNF-α and palmitate through TLR4/IRF3, and c-Jun/NF κB signaling pathway. Overall, the manuscript is well-written. Some concerns described below.
Point 1: In Fig.5, Western blotting showed high level of p-NF-κB in TNF-α-treated cells. Why the expression of MIP-1 α protein decreased scarcely in TNF-α-treated cells with resveratrol?
Response 1: In the “classical pathway”, NF-κB is engaged by activation of cell surface receptors such as TLRs, IL-1R, and TNFR. In MyD88-dependent TLR signaling pathway, TLR recruits TIR domain-containing adaptor protein in cell membrane recruits MyD88, triggering the early-phase NF-κB activation. In TRIF-dependent TLR signaling pathway, TLR4 forms a signaling complex with TRAM and TRIF, resulting in the late-phase NF-κB activation via IRF3 engagement and IFN-β transcription. IKK phosphorylates IκBα and IκBβ, leading to polyubiquitination and degradation of IκBs via 26S proteasome and release of NF-κB which translocates to nucleus to activate target gene expression (1, 2). In the “alternative pathway”, mainly involved in immune response, TNFR activation initiates the NIK to stimulate IKKα-induced phosphorylation and processing of NF-κB2 precursor. Activated NF-κB2, NIK, and RelB form a complex which translocates to nucleus to activate target gene expression (3, 4).
Emerging evidence shows that resveratrol plays potential therapeutic roles in obesity, diabetes, cancer, respiratory diseases, cardiovascular disease, ischemic injury, sarcopenia, and neurodegeneration. Resveratrol has multiple pharmacologic effects such as anti-inflammatory, anti-oxidative, anti-diabetic, anti-cancer, anti-depression, cardio- and neuro-protective as well as regulation of metabolism (5). Biological activity and functions of resveratrol are mediated via NF-κB, MAPK, and other pathways; especially, inhibition of the NF-κB signaling pathway by resveratrol plays a critical role in treating various pathological conditions. Resveratrol-mediated suppression of several cytokines or enzymes such as TNF-α, IL-1β, IL-6, TGF-β, MIP, MMP-9, and COX-2 indicates that resveratrol mediates these anti-inflammatory effects by most likely inhibiting the non-canonical pathway of NF-κB activation.
The western blot shows increased phosphorylation of c-Jun and NF-κB proteins in THP-1 cells co-stimulated with palmitate and TNF-α compared to those stimulated with either palmitate or TNF-α, indicating that a potent cooperative signaling was induced by co-stimulation.
Point 2: In Fig. 6, the effect of 10 mM H2O2 (high dosage) on MIP-1α induction in TNF-α and palmitate co-treated cells was not obvious. In my opinion, it was inappropriate to state that “ Oxidative stress further promotes the MIP-1α expression co-induced by palmitate and TNF-α “ . In addition, the role of ROS in the synergistic induction of MIP-1α in the combined treatment needed to be discussed.
Response 2: It is true that MIP-1α mRNA expression differs non-significantly between “palmitate+TNF-α” treatments with or without H2O2, whereas there is a significant difference between treatments regarding MIP-1α secreted protein expression (P=0.002; Fig 6C). It remains unclear whether the ROS-induced changes in MIP-1α mRNA stability and/or protein turnover rates might be the reason for this disparity between gene and protein expression; albeit, significantly enhanced MIP-1α protein secretion by the cells supports its induction/upregulation by oxidative stress. In keeping with better clarity, the statement in question has been re-phrased as follows: “Oxidative stress promotes the MIP-1α protein secretion co-induced by palmitate and TNF-α”.
Submission Date
04 June 2020
Date of this review
19 Jun 2020 02:15:08
References
- T. Kawai, S. Akira, The role of pattern-recognition receptors in innate immunity: update on Toll-like receptors. Nature immunology 11, 373-384 (2010).
- E. O'Dea, A. Hoffmann, The regulatory logic of the NF-kappaB signaling system. Cold Spring Harb Perspect Biol 2, a000216-a000216 (2010).
- A. R. Brasier, The NF-κB regulatory network. Cardiovascular Toxicology 6, 111-130 (2006).
- E. Zandi, Y. Chen, M. Karin, Direct phosphorylation of IkappaB by IKKalpha and IKKbeta: discrimination between free and NF-kappaB-bound substrate. Science 281, 1360-1363 (1998).
- L. Xu, B. O. A. Botchway, S. Zhang, J. Zhou, X. Liu, Inhibition of NF-κB Signaling Pathway by Resveratrol Improves Spinal Cord Injury. Frontiers in Neuroscience 12, (2018).

Reviewer 2 Report
Summary:
The authors are investigating the MIP-1a expression in human monocytes/macrophages in response to a TNFa and/or palmitate stimulation.
In response to TNFa and palmitate the cells are producing significantly more MIP-1a at a transcriptional and translational level. Moreover, the authors show that this production is lowered after a TLR4 inhibition or blockade. Inhibiting IRF3, a downstream protein of TLR4, could reproduce the results of the TLR4 inhibition/blockade.
The authors also showed that ROS alone could induce a MIP-1a production and enhance its expression after either a palmitate stimulation or a TNFa stimulation. Interestingly ROS could not increase the MIP-1a secretion after a TNFa/palmitate co-stimulation.
Comments:
- Throughout most figures: upon inhibition (siRNA, Ab or chemical), MIP1a shows still a significantly induced expression upon double stimulation compared to single treatments (yet at lower level). Thus, it cannot be fully excluded that alternative pathways contribute to the induction of MIP1a or if it is a result of incomplete inhibition (after e.g. siRNA or inhibitor treatment). Thus, I would suggest to choose a more careful wording throughout the text (instead of “abrogated” e.g. “reduced”).
- H2O2 treatments: The H2O2 data is not convincing in its current form.
- The heading in 3.5 is not supported by the data presented in Figure 6B. According to the title, an “further” expression upon Palm+TNF+H2O2 compared to Palm+TNF would be expected. Even though there seems to be significant increase for protein level in Figure 6C, it is overall not convincing (see also comment on statistics in point 8).
- In regard to the previous comment, ROS scavenging should result in reduced expression upon Palm+TNF activation. This data is so far lacking and a respective conclusion cannot be done based on the data provided. Figure 7 shows that ROS scavengers can reduce H2O2 effects. Thus, the authors demonstrate that ROS alone can trigger MIP1a (independent of TNFR and/or TLR4 activation). But these results are not related to the key question of this study: if ROS promotes MIP1a expression in context of Palm+TNF activation. As it is known that activation of these receptors triggers inflammation and that inflammation increases intracellular ROS, it could be possible that ROS acts as a kind of downstream effector to activate MIP1a (because H2O2 alone can increase MIP1a) expression. In the current version of this manuscript the relation between ROS, TLR4 and TNFR is not clearly described and not clearly supported by the experimental data. At first reading of the manuscript it was not clear to me how the data of Figure 7 contribute to the manuscript. A rewriting of this section could help here to increase clarity (ROS alone induces MIP1a as the main conclusion of it).
- Based on the M+M section different incubation times have been applied throughout the manuscript. Thus, it would help the reader to clearly indicate the incubation times directly within each figure legend rather than referencing to the M+M section.
- 10mM H2O2 treatment appears very high to me as other mammalian cells die already at concentration around 1mM. I am wondering if such a high concentration is physiologically relevant. Can the authors provide data on viability for THP1 cells exposed to 10mM H2O2 for 10h? Also for the evaluation of the data in figure 7 it would be helpful to measure ROS levels after TLR4 and/or TNFR activation. àHow do the H2O2 induced ROS levels relate to endogenous ROS levels during inflammation (physiological relevance)?
- Even though as referenced in the introduction, Palmitate seems to be a non-canonical TLR4 activator. Thus, it would be of interest to test the effect of palmitate on MIP1a expression in comparison to a classical TLR4 activation (e.g. LPS) and vice versa: the effect of palmitate on classical TLR4 related cytokines (e.g. IL1b, TNFa, IL6).
- Can the authors comment on the 10-fold difference in Y-axis in figure 2D vs 2F (applies also to other data shown throughout the manuscript). Also when comparing gene expression with respective protein levels (2C, D and 2E, F) there seems to be an inverse correlation).
- Figure 4A, B is lacking a “WT” control. As the data is presented now the statement cannot be fully supported. Commonly, when working with a knock-down or knock-out cell line the respective control (e.g. THP wild type in this case) should be included (as also done in all other panels presented. In fact, looking at the data and the values presented on the Y-axis and comparing those to figure 1a, MyD88 could act as a repressor on MIP1a, because MIP1a expression looks increased compared to THP1 WT. However, to allow a proper comparison these cell lines should be analyzed in one experiment together. As further explanation to this observation: if we only compare IRF3siRNA (Figure 4c) alone, the relative changes would look similar as shown in Figure 4a (increased expression of MIP1a upon double stimulation compared to single stimulation). However, the conclusion is completely different.
- Figure 5c: the Western Blot is not showing the Blot for the total protein (c-Jun and NFkB).
- For all FACS data presented, a respective quantification (presented as bar plots) would be helpful.
- Statistics:
As indicated in the figure legends all experiments have been repeated three times. However, if I understand correctly, the presented data show one representative experiment (and thus technical replicates). Can the authors please state how the statistical calculation has been performed? Was this on one experiment with technical triplicates (n = 1) or on all three independent experiments (n = 3).
Author Response
Response to Reviewer 2 Comments
Comments and Suggestions for Authors
Summary:
The authors are investigating the MIP-1a expression in human monocytes/macrophages in response to a TNF-a and/or palmitate stimulation.
In response to TNF-a and palmitate the cells are producing significantly more MIP-1a at a transcriptional and translational level. Moreover, the authors show that this production is lowered after a TLR4 inhibition or blockade. Inhibiting IRF3, a downstream protein of TLR4, could reproduce the results of the TLR4 inhibition/blockade.
The authors also showed that ROS alone could induce a MIP-1a production and enhance its expression after either a palmitate stimulation or a TNF-a stimulation. Interestingly ROS could not increase the MIP-1a secretion after a TNF-a/palmitate co-stimulation.
Comments:
Point 1: Throughout most figures: upon inhibition (siRNA, Ab or chemical), MIP1a shows still a significantly induced expression upon double stimulation compared to single treatments (yet at lower level). Thus, it cannot be fully excluded that alternative pathways contribute to the induction of MIP1a or if it is a result of incomplete inhibition (after e.g. siRNA or inhibitor treatment). Thus, I would suggest to choose a more careful wording throughout the text (instead of “abrogated” e.g. “reduced”).
Response 1: Thanks for pointing out to this inadvertent mistake. As suggested, the word “abrogated” has been replaced with “reduced” throughout the MS text.
Point 2: H2O2 treatments: The H2O2 data is not convincing in its current form:
(i) The heading in 3.5 is not supported by the data presented in Figure 6B. According to the title, an “further” expression upon Palm+TNF+H2O2 compared to Palm+TNF would be expected. Even though there seems to be significant increase for protein level in Figure 6C, it is overall not convincing (see also comment on statistics in point 8)
Response 2 (i): It is true that MIP-1α mRNA expression differs non-significantly between “palmitate+TNF-α” treatments with or without H2O2, however, there is a significant difference between treatments regarding MIP-1α secreted protein expression (P=0.002; Fig 6C). It remains unclear whether the ROS-induced changes in MIP-1α mRNA stability and/or turnover rates might be the reason for this disparity between gene and protein expression; albeit, significantly enhanced MIP-1α protein secretion by the cells supports MIP-1α induction/upregulation by oxidative stress. In order to improve clarity, the statement in question has been re-phrased as follows: “MIP-1α expression induced by palmitate and/or TNF-α, in presence or absence of oxidative stress” (lines 452-453).
(ii) In regard to the previous comment, ROS scavenging should result in reduced expression upon Palm+TNF activation. This data is so far lacking, and a respective conclusion cannot be done based on the data provided. Figure 7 shows that ROS scavengers can reduce H2O2 effects. Thus, the authors demonstrate that ROS alone can trigger MIP1a (independent of TNFR and/or TLR4 activation). But these results are not related to the key question of this study: if ROS promotes MIP1a expression in context of Palm+TNF activation. As it is known that activation of these receptors triggers inflammation and that inflammation increases intracellular ROS, it could be possible that ROS acts as a kind of downstream effector to activate MIP1a (because H2O2 alone can increase MIP1a) expression. In the current version of this manuscript the relation between ROS, TLR4 and TNFR is not clearly described and not clearly supported by the experimental data. At first reading of the manuscript it was not clear to me how the data of Figure 7 contribute to the manuscript. A rewriting of this section could help here to increase clarity (ROS alone induces MIP1a as the main conclusion of it).
Response 2 (ii): The reviewer’s comment that ROS per se drives MIP-1α protein secretion is absolutely true and Fig 7 clearly shows that H2O2-induced oxidative stress enhances MIP-1α protein secretion via a ROS-dependent mechanism because treatments with anti-oxidants or ROS scavengers significantly reduce the secretion of MIP-1α protein. As of the question whether or not palmitate or TNF-α treatments might also induce ROS, we have added new data showing significant increase in the intracellular ROS following treatments with palmitate (P=0.001), TNF-α (P=0.006), and palmitate+TNF-α (P=0.0008) as compared to vehicle-treated control (Figs 6B-E). In the “alternative or non-canonical signaling pathway” of NF-κB activation (mainly involved in immune responses), TNFR engagement and activation initiates the NF-κB-inducing kinase (NIK; also called MAP3K14) to stimulate IKKα-induced phosphorylation and processing of NF-κB2 precursor. Activated NF-κB2, NIK, and RelB form a complex which translocates to nucleus to activate target gene expression (3, 4). Inflammatory signaling pathways leading to NF-κB activation are influenced by ROS, and growing evidence now supports that TNF-α and ROS influence each other in a positive feedback loop (6, 7). Similarly, exposure to saturated fatty acid palmitate also led to ROS overproduction in rat pancreatic beta cells (8). These studies are consistent with our data showing ROS induction by both palmitate and TNF-α. As expected, level of ROS induction is even higher when cells are co-stimulated with palmitate+TNF-α. Not surprisingly, MIP-1α protein secretion is higher when cells are co-stimulated with palmitate+TNF-α in presence of H2O2 compared to co-stimulation without H2O2 (P=0.002; Fig 6G). Nonetheless, the discordance between MIP-1α protein and mRNA could possibly be due to ROS-induced changes in steady state levels/half-lives of mRNAs, altered protein turnover rates and translational reprogramming when the cellular stress is induced by H2O2 treatment, offsetting the ROS’ homeostatic regulation (9, 10).
(iii) Based on the M+M section different incubation times have been applied throughout the manuscript. Thus, it would help the reader to clearly indicate the incubation times directly within each figure legend rather than referencing to the M+M section.
Response 2 (iii): Modified as advised.
(iv) 10mM H2O2 treatment appears very high to me as other mammalian cells die already at concentration around 1mM. I am wondering if such a high concentration is physiologically relevant. Can the authors provide data on viability for THP1 cells exposed to 10mM H2O2 for 10h? Also for the evaluation of the data in figure 7 it would be helpful to measure ROS levels after TLR4 and/or TNFR activation. How do the H2O2 induced ROS levels relate to endogenous ROS levels during inflammation (physiological relevance)?
Response 2 (iv): Our past data show that THP-1 monocytic cells are well tolerant to H2O2 treatment and concentrations of 10mM (10h) do not cause significant toxicity in these cells. The viability assessment by trypan blue dye exclusion assay in triplicate shows non-significant change in the mean cell viability from 91% to 87% over a time period of 10h compared to vehicle-treated control (P=0.458). The viability data are now included in the supplementary information.
Also, as advised, ROS measurements in cells following TLR4 and/or TNFR activation have been added. As expected, the maximum ROS induction was observed with palmitate+TNF-α co-stimulation (Mean Staining Index: 33.87), followed in order by palmitate alone (Mean Staining Index: 5.89) and TNF-α alone (Mean Staining Index: 2.74), all of which differed significantly from control. Please see new Fig 6E.
Since H2O2 is the most stable and longest-lived species among ROS, as well as it acts as both an intracellular second messenger and an intercellular messenger (11-13); THP-1 monocytic cells were treated with H2O2 to examine effects of ROS on MIP-1α protein expression. When THP-1 cells are exposed to H2O2, ROS is produced and used as second messengers in response to extrinsic stimuli. Besides, monocytes/macrophages also intrinsically produce endogenous ROS, mainly H2O2, resulting from the activity of cell membrane NADPH oxidases (NOX pathway) and during the mitochondrial oxidative phosphorylation and P450 systems (14). Whereas, endogenous ROS is counteracted and homeostatically regulated by antioxidant enzymes such as superoxide dismutase, catalase, and glutathione peroxidase, exogenous ROS exceeds this homeostatic threshold and affects the expression of various genes. Consistent with our observation of ROS induction by pro-inflammatory stimuli including palmitate, LPS, and TNF-α, others have also demonstrated ROS induction by TNF-α, IL-1β, IFN-γ, IL-6, IL-8, and LPS (15-17). ROS plays role in both physiological and pathophysiological signal transduction. Accumulating evidence supports that ROS levels are upregulated during metabolic and inflammatory conditions (18, 19).
As of 10mM concentration of H2O2 used in our study to induce oxidative stress/ROS in THP-1 monocytic cells, it was most likely greater than the intrinsic H2O2 among endogenous ROS; however, it was physiologically well tolerated and no significant cytotoxicity or cell death occurred over a time period of 10h (data are included as Supplementary Fig 1). This H2O2 concentration was found to optimally induce the MIP-1α expression in THP-1 monocytic cells.
Point 3: Even though as referenced in the introduction, Palmitate seems to be a non-canonical TLR4 activator. Thus, it would be of interest to test the effect of palmitate on MIP1a expression in comparison to a classical TLR4 activation (e.g. LPS) and vice versa: the effect of palmitate on classical TLR4 related cytokines (e.g. IL1b, TNFa, IL6).
Response 3: In view of a comparison between canonical and non-canonical TLR4 activation, we have added new data (Supplementary Fig S4A) showing MIP-1α gene and protein expression in response to palmitate, TNF-α, Palmitate+TNF-α as well as LPS, and LPS+TNF-α. Next, in order to further demonstrate effects of palmitate stimulation on a classical TLR4-related cytokine, we have added new data (Supplementary Fig S4B) showing IL-6 expression in response to afore-mentioned cell stimulations. Briefly, our data show that canonical TLR4 activation for MIP-1α expression is more potent than non-canonical TLR4 activation. On the other hand, palmitate is less potent in inducing a classical TLR4 related cytokine IL-6 than its induction by LPS.
Point 4: Can the authors comment on the 10-fold difference in Y-axis in figure 2D vs 2F (applies also to other data shown throughout the manuscript). Also, when comparing gene expression with respective protein levels (2C, D and 2E, F) there seems to be an inverse correlation).
Response 4: We have found that various treatments may change the cellular responsiveness to agonistic stimuli; therefore, it would not be rational to compare MIP-1α expression data among Figs 2A-B, 2C-D, and 2E-F as the cells have been subjected to different treatments i.e. TLR4 siRNA transfection, labeling with TLR4 neutralizing Ab, and treatment with TLR4 chemical inhibitor OxPAPC, in respective order. Such data ought to be compared with the respective control within each type of treatment/experiment itself. Notably, the data presented are highly reproducible and represent the values obtained from three independent experiments; and in each experiment, every treatment was done in triplicate wells.
Point 5: Figure 4A, B is lacking a “WT” control. As the data is presented now the statement cannot be fully supported. Commonly, when working with a knock-down or knock-out cell line the respective control (e.g. THP wild type in this case) should be included (as also done in all other panels presented. In fact, looking at the data and the values presented on the Y-axis and comparing those to figure 1a, MyD88 could act as a repressor on MIP1a, because MIP1a expression looks increased compared to THP1 WT. However, to allow a proper comparison, these cell lines should be analyzed in one experiment together. As further explanation to this observation: if we only compare IRF3siRNA (Figure 4c) alone, the relative changes would look similar as shown in Figure 4a (increased expression of MIP1a upon double stimulation compared to single stimulation). However, the conclusion is completely different.
Response 5: THP-1-XBlueTM-defMyD cells (MyD88-/- THP-1 cells; purchased from InvivoGen) are derived from the human monocytic THP-1 cells and these cells stably express an NF-κB and AP-1 inducible secreted embryonic alkaline phosphatase (SEAP) reporter gene; however, these cells are deficient in MyD88 activity and thus do not respond to agonists of TLR2/6 (FSL-1), TLR4 (LPS), and TLR8 (CL075), all of which induce MyD88-dependent signaling. MyD88 deficiency in THP-1-XBlueTM-defMyD cells was confirmed by qRT-PCR. THP-1-XBlueTM-defMyD cells are resistant to the selectable markers hygromycin B and Zeocin which are added to complete RPMI (HygroGold 100µg/ml; Zeocin 200ug/ml) to maintain plasmids’ expression. In our lab, we have already tested the THP-1-XBlueTM-defMyD cells for non-responsiveness to MyD88-signaling agonists FSL-1, LPS, and CL075 but intact responsiveness to MyD88-independent receptor agonists such as TNF-α (TNFR ligand) and Tri-DAP (NOD1 ligand), compared to a positive control cell line THP-1-XBlueTM cells which are fully efficient for MyD88 activity. Last but not least, we do agree that the inclusion of this positive control cell line in the experiment would have further substantiated the data presented.
Point 6: Figure 5c: The Western Blot is not showing the Blot for the total protein (c-Jun and NFkB).
Response 6: Western blots have been redone, showing expression of both phosphorylated and total c-Jun and NF-κB.
Point 7: For all FACS data presented, a respective quantification (presented as bar plots) would be helpful.
Response 7: Done as advised.
Point 8: Statistics: As indicated in the figure legends all experiments have been repeated three times. However, if I understand correctly, the presented data show one representative experiment (and thus technical replicates). Can the authors please state how the statistical calculation has been performed? Was this on one experiment with technical triplicates (n = 1) or on all three independent experiments (n = 3).
Response 8: The data represent mean±SEM from three independent experiments (n=3); treatments in each experiment were performed in triplicate wells.
Submission Date
04 June 2020
Date of this review
18 Jun 2020 16:27:25
References
- A. R. Brasier, The NF-κB regulatory network. Cardiovascular Toxicology 6, 111-130 (2006).
- E. Zandi, Y. Chen, M. Karin, Direct phosphorylation of IkappaB by IKKalpha and IKKbeta: discrimination between free and NF-kappaB-bound substrate. Science 281, 1360-1363 (1998).
- L. Xu, B. O. A. Botchway, S. Zhang, J. Zhou, X. Liu, Inhibition of NF-κB Signaling Pathway by Resveratrol Improves Spinal Cord Injury. Frontiers in Neuroscience 12, (2018).
- H. Blaser, C. Dostert, T. W. Mak, D. Brenner, TNF and ROS Crosstalk in Inflammation. Trends Cell Biol 26, 249-261 (2016).
- R. Fischer, O. Maier, Interrelation of oxidative stress and inflammation in neurodegenerative disease: role of TNF. Oxid Med Cell Longev 2015, 610813 (2015).
- Y. Sato et al., Palmitate induces reactive oxygen species production and β-cell dysfunction by activating nicotinamide adenine dinucleotide phosphate oxidase through Src signaling. J Diabetes Investig 5, 19-26 (2014).
- C. M. Grant, Regulation of translation by hydrogen peroxide. Antioxidants & redox signaling 15, 191-203 (2011).
- P. Brenneisen, K. Briviba, M. Wlaschek, J. Wenk, K. scharffetter-kochanek, Hydrogen peroxide (H2O2) Increases the Steady-State mRNA Levels of Collagenase/MMP-1 in Human dermal Fibroblasts. Free radical biology & medicine 22, 515-524 (1997).
- C. Li, R. M. Jackson, Reactive species mechanisms of cellular hypoxia-reoxygenation injury. Am J Physiol Cell Physiol 282, C227-241 (2002).
- S. J. Weiss, Oxygen, ischemia and inflammation. Acta Physiol Scand Suppl 548, 9-37 (1986).
- J. M. Servitja, R. Masgrau, R. Pardo, E. Sarri, F. Picatoste, Effects of oxidative stress on phospholipid signaling in rat cultured astrocytes and brain slices. J Neurochem 75, 788-794 (2000).
- C. Murdoch, M. Muthana, C. E. Lewis, Hypoxia regulates macrophage functions in inflammation. J Immunol 175, 6257-6263 (2005).
- H. Liu, R. Colavitti, Rovira, II, T. Finkel, Redox-dependent transcriptional regulation. Circ Res 97, 967-974 (2005).
- K. Sverrisson, J. Axelsson, A. Rippe, D. Asgeirsson, B. Rippe, Acute reactive oxygen species (ROS)-dependent effects of IL-1beta, TNF-alpha, and IL-6 on the glomerular filtration barrier (GFB) in vivo. Am J Physiol Renal Physiol 309, F800-806 (2015).
- D. Yang et al., Pro-inflammatory cytokines increase reactive oxygen species through mitochondria and NADPH oxidase in cultured RPE cells. Exp Eye Res 85, 462-472 (2007).
- S. J. Forrester, D. S. Kikuchi, M. S. Hernandes, Q. Xu, K. K. Griendling, Reactive Oxygen Species in Metabolic and Inflammatory Signaling. Circ Res 122, 877-902 (2018).
- M. A. Chelombitko, Role of Reactive Oxygen Species in Inflammation: A Minireview. Moscow University Biological Sciences Bulletin 73, 199-202 (2018).
- R. Ahmad et al., The Synergy between Palmitate and TNF-α for CCL2 Production Is Dependent on the TRIF/IRF3 Pathway: Implications for Metabolic Inflammation. The Journal of Immunology, ji1701552 (2018).
- A. Hasan et al., TNF-α in Combination with Palmitate Enhances IL-8 Production via The MyD88- Independent TLR4 Signaling Pathway: Potential Relevance to Metabolic Inflammation. International journal of molecular sciences 20, 4112 (2019).

Reviewer 3 Report
The authors conducted experimental studies investigating the effect of co-stimulation of TNF-alpha and palmitate on MIP-1alpha expression and secretion in human THP-1 and primary monocytes. Standard and acceptable methods of qRT-PCR, Western blots, ELISA, and flow cytometry were used.
Strengths:
This study uses multiple methods and gene deletion cell lines or experimental techniques to determine mediating effects. Methods section was detailed to be able to assess scientific merit.
Weaknesses:
This study does provide detailed cellular effects and potential mechanisms of TNF-alpha, palmitate, and its co-stimulation, but there are a few major weaknesses and minor weaknesses that need to be addressed.
Major weakness
- The major weakness is the use of resveratrol as the NFkappaB inhibitor. Resveratrol has many targets in cells including AMPK, mTOR, and other signaling mediators and also is a direct scavenger of reactive oxygen species. I do not consider it a NFKappaB specific inhibitor. I suggest moving the resveratrol data the ROS scavenging section with the other phytochemical (curcumin) and the NFKappaB western blot in with your conclusions about the IRF3 pathway. Conclusions about NFkappaB need to also be modified throughout the paper.
Minor weaknesses
- Justify dosages for TNF-alpha and palmitate (i.e. based on previous experiments, physiological relevance)
- Fold change is singular.
- Suggest using control or vehicle control instead of mock.
- In discussion
- Elaborate on possible cellular sources of ROS in your model/figure.
- Differentiate between ROS added in the experiments and physiological ROS in your figure.
- Elaborate on the impact of your findings and future directions.
Author Response
Response to Reviewer 3 Comments
Comments and Suggestions for Authors
The authors conducted experimental studies investigating the effect of co-stimulation of TNF-alpha and palmitate on MIP-1alpha expression and secretion in human THP-1 and primary monocytes. Standard and acceptable methods of qRT-PCR, Western blots, ELISA, and flow cytometry were used.
Strengths:
This study uses multiple methods and gene deletion cell lines or experimental techniques to determine mediating effects. Methods section was detailed to be able to assess scientific merit.
Weaknesses:
This study does provide detailed cellular effects and potential mechanisms of TNF-alpha, palmitate, and its co-stimulation, but there are a few major weaknesses and minor weaknesses that need to be addressed.
Major weakness
Point 1: The major weakness is the use of resveratrol as the NFkappaB inhibitor. Resveratrol has many targets in cells including AMPK, mTOR, and other signaling mediators and also is a direct scavenger of reactive oxygen species. I do not consider it a NFKappaB specific inhibitor. I suggest moving the resveratrol data the ROS scavenging section with the other phytochemical (curcumin) and the NFKappaB western blot in with your conclusions about the IRF3 pathway. Conclusions about NFkappaB need to also be modified throughout the paper.
Response 1: Thanks for kind suggestions. Modifications have been accordingly done.
Minor weaknesses
Point 2: Justify dosages for TNF-alpha and palmitate (i.e. based on previous experiments, physiological relevance)
Response 2: TNF-α and palmitate were used in standard concentrations as referred to in previous publications (20, 21).
Point 3: Fold change is singular.
Response 3: Corrected throughout the text as pointed out.
Point 4: Suggest using control or vehicle control instead of mock.
Response 4: Corrected throughout the Figs as pointed out.
Point 5: In discussion:
(i) Elaborate on possible cellular sources of ROS in your model/figure.
Response 5 (i): ROS in living cells are produced during the activity of various cellular systems that are localized in the: (1) plasma membrane (mainly generated by activities of membrane-bound NADPH oxidases/NOXs, as well as during synthesis of arachidonic acid metabolites including prostaglandins, leukotrienes, and thromboxanes by activities of cyclooxygenase/COX and lipoxygenase/LOX); (2) cytosol (by catalytic activity of xanthine oxidase/XO); (3) mitochondria (during oxidative phosphorylation for ATP synthesis, by activities of super oxide dismutase/SOD and oxidoreductases); (4) endoplasmic reticulum (i - during xenobiotic metabolism by activities of microsomal cytochrome P450-dependent monooxygenase system; ii - during synthesis of unsaturated fatty acids by activities of NADH cytochrome b5 reductase and desaturases; & iii - during protein folding by activities of protein disulfide isomerase/PDI); (5) peroxisomes (during fatty acid β- and α-oxidation as well as amino acid and glyoxylate metabolism etc. by activities of iNOS, peroxisomal and xanthine oxidases); and (6) lysosomes (during terminal degradation of macromolecules by activities of acidic hydrolases). The cellular ROS generated by these diverse sources are tightly regulated by homeostatic mechanisms involving anti-oxidative enzymes and this homeostasis is impaired under conditions of cellular stress, senescence, and disease. The suggested model/Fig in our paper is now modified accordingly to show different sources of endogenous ROS.
(ii) Differentiate between ROS added in the experiments and physiological ROS in your figure.
Response 5 (ii): Done as advised.
(iii) Elaborate on the impact of your findings and future directions.
Response 5 (iii): Done as advised.
Submission Date
04 June 2020
Date of this review
19 Jun 2020 21:24:35

Round 2
Reviewer 1 Report
The revised manuscript has been corrected according to comments. In my opinion, it will be considered to be accepted for publication.